



# Evaluation of tropical water vapour from CMIP6 GCMs using the ESA CCI "Water Vapour" climate data records

Jia He[1], Helene Brogniez[1], and Laurence Picon[2]

[1]LATMOS/IPSL, UVSQ Université Paris-Saclay, CNRS, Guyancourt, France
[2]LMD/IPSL, Sorbonne-Université, CNRS, Paris, France

**Correspondence:** Jia He (jia.he@latmos.ipsl.fr) Helene Brogniez (helene.brogniez@latmos.ipsl.fr)

**Abstract.** The tropospheric water vapour data record generated within the ESA Climate Change Initiative "Water Vapour" project (ESA TCWV-COMBI) is used to evaluate the interannual variability of global climate models (CMIP6 framework under AMIP scenarios) and reanalysis (ECMWF ERA5). The study focuses on the tropical belt, with a separation of oceanic and continental situations. The intercomparison is performed according to the probability density function (PDF) of the total

column water vapour (TCWV) defined yearly from the daily scale, as well as the evolution of the large-scale overturning circulation. The observational diagnostic relies on the decomposition of the tropical atmosphere into percentile of the PDF and into dynamical regimes defined from the atmospheric vertical velocity. Large variations are observed in the patterns among the data records, especially over tropical-land, while oceanic situations show more similarities in both interannual variations and percentile extremes. The signatures of El Nino/La Nina events, driven by the sea surface temperatures, are obvious over

the oceans. Differences also occur over land for both trends (a strong moistening is observed in the ESA TCWV-COMBI data record which is absent of CMIP6 models and ERA5) and extremes years. The discrepancies are probably associated with the scene selection applied in the data process. Other sources of differences, linked to the models and their parametrizations, are highlighted.

## 1 Introduction

Water vapour is short-lived yet sufficiently abundant component of the atmosphere that has both direct and indirect impact on weather and environment. It is one of the most important greenhouse gases and it plays an critical role in the hydrological cycle and climate system (Held and Soden, 2000). It is a radiatively important atmospheric constituent that influences atmospheric energy exchange through interactions with solar and thermal radiations (Raval and Ramanathan, 1989) with strong positive feedbacks (Sherwood et al., 2010). The precipitable water, mainly concentrated in the atmospheric boundary layer, is directly

influenced by the surface temperature from robust thermodynamical constraints which predicts a global increase in the concentration of the boundary layer water vapour close to 7% /°C, confirmed by simulations and observations (Allan et al., 2014). Overall an increase in atmospheric moisture drives amplifying feedbacks, yielding to changes in evaporation and precipitation patterns at a global scale and to amplify heavy precipitation events (Hartmann et al., 2013; Allan et al., 2020).
Whether it is for weather forecasting, for understanding the evolution of cloud cells or for climate change studies, observing



the distribution of atmospheric water vapour at any point in the atmosphere is a central issue. That is why the Global Climate

Observing System (GCOS) declared the atmospheric water vapour as one of the Essential Climate Variables (ECVs) (GCOS,

2016). However, there are almost 5 orders of magnitude on the water vapour concentration between the surface and the top of

the meteorological atmosphere. Evapotranspiration over continental surfaces, atmospheric dynamics or cloud formation rein-

force the horizontal and vertical gradients. Unfortunately, the required accuracy at all spatial and temporal scales is difficult

to achieve, which leads to the combination of different types of measurements, each with their strengths and weaknesses, de-

pending on the objectives (Wulfmeyer et al., 2015).

Satellite observations provide global water vapour measurements since the 1970s. Current sensors allow to observe the spatial

distribution of water vapour according to three quantities: the vertical profile of specific humidity (q, in kg/kg), the relative

humidity of the upper troposphere (UTH, in %) and the total water vapour content (TCWV, in kg/m2). These sensors are

deployed either in polar orbit, geostationary orbit or inclined orbit to enhance the temporal sampling of a latitude band. Many

programs have been developed to get long-term observations with high spatial and temporal resolutions. Among them, the

European Space Agency (ESA) Climate Change Initiative (CCI) program has been created to explore the full potential of its

Earth Observation missions and to generate climate data record (CDR) associated to each of the 23 ECVs. Among the ESA

CCI projects, the CCI Water Vapour project (hereafter CCI_WV, https://climate.esa.int/en/projects/water-vapour) started in

2018 with the objectives to generate long-term coherent datasets of tropospheric and stratospheric water vapour.

The tropical belt (30°S-30°N) is a pivotal region in the Earth's climate. In this part of the globe, regional variations of the

hydrological cycle are closely related to the Hadley-Walker overturning cells, that define the tropospheric circulation, and to

its long-term changes (such as its observed slowdown and poleward expansion (Ma et al., 2018; Lu et al., 2007)).

Studies linking tropical climate evolution and water vapour distribution are done with both models and long-term observations.

For instance, the infrared (IR) observations from the High resolution Infrared Radiation Sounder (HIRS) instrument have been

used to investigate the interannual variability of tropical moisture of an atmospheric model (Huang et al., 2005). Following the

same approach Chung et al. (2011) have analyzed the variability of the simulated UTH from a climate model using both IR and

microwave measurements and showed that the wet bias of the model was related to errors in simulating the intensity of large-

scale circulation. Globally, improvements have been noticed in the simulation of the tropospheric water vapour distribution

and variability between the Coupled Model Intercomparison Project (CMIP5, release in 2014) and the earlier CMIP3 excercise

(release in 2010) (Jiang et al., 2012). However while the models perform best for the boundary layers over the oceans, most

likely thanks to the thermodynamic constraint imposed by the sea surface temperature, strong biases remain in the upper layers

of the troposphere, where clouds and atmospheric dynamics have large uncertainties.

A recent intercomparison work of a selection of available long-term datasets has been conducted under the auspices of the

Global Energy and Water Exchanges (GEWEX) program of the world climate research programme (WCRP) (Schröder et al.,

2019). This intercomparison has highlighted both the complementarity between the sensors but it has also underlined the

caveats in the studies of trends and variabilities induced by artificial break points contained in the CDRs such as calibration

changes, retrieval algorithms, resolution changes that impact the sampling, etc.. Apart from these important points that are

inherent to the development of robust time series suitable to investigate climate variability, the evaluation of the distribution





and variability of the water vapour with respect to large-scale circulation is still of upmost importance. The last IPCC assessment report (AR6, 2021) contains an entire chapter on the water cycle and this chapter highlights the role of the large-scale atmospheric circulation in the driving of regional changes of atmospheric moisture fluxes and in the position and strength of the tropical rain belt (Arias et al., 2021).

The present work follows the previous analysis framework proposed by Bony et al. (2004) and investigate the the variabiltity of total column water vapour (TCWV) of a selection of Global Climate Models (GCMs) that have participated to the CMIP6 exercise (Eyring et al., 2016). The ESA CCI_WV CDRs are studied from the aspect of the large-scale circulation. Section 2 describes the various datasets : the ESA CCI_WV CDRs, the CMIP6 GCMs and the ERA5 reanalysis. Section 3 is dedicated to the method of evaluation itself, while Section 4 discusses the results. Finally the conclusions are drawn in Section 5.

## 2 Data Description

We focus on the tropical belt (30°S - 30°N). Data obtained from ESA CCI_WV, CMIP6 GCMs, and ERA5 reanalysis are studied at daily scale and over the period 2003 - 2014. The monthly atmospheric vertical velocity at 500hPa ($\omega 500$) of each data record is used as a proxy of large-scale circulation following Bony et al. (2004). Note that the vertical velocity from ERA5 is also used as the dynamical reference for the CCI_WV CDRs.

### 2.1 ESA CCI_WV Climate Data Records

The Phase 1 of the ESA CCI_WV project was dedicated to built climate records of both tropospheric and stratospheric water vapour. The project provides daily and monthly water vapour observations on the global scale with spatial resolution of 0.5 and 0.05 degree for the period of 2002-2017.

The TCWV is commonly defined as the vertically integrated water vapour over the full column with units of kg/m$^2$. Observations from microwave (MW) imagers (namely SSM/I, SSMIS, AMSR-E and TMI) over the ice-free ocean, partly based on a fundamental CDR (Fennig et al., 2020), and near-infrared (NIR) imagers (including MERIS, MODIS-Terra and OLCI) over land, coastal ocean and sea-ice have been combined within the ESA CCI_WV project. Details of the retrieval are discussed in Andersson et al. (2010) and Schröder et al. (2013) for the MW imagers. The algorithm for NIR imagers are discussed in Lindstrot et al. (2012), Diedrich et al. (2015) and Preusker et al. (2021). The MW and NIR data streams are processed independently and combined afterwards so that the individual TCWV values and their uncertainties remain unchanged. The available spatial resolutions of the combined data record (hereafter TCWV-COMBI) are 0.5°x0.5° and 0.05°x0.05°, where the NIR based data are averaged and the MW-based data are oversampled to produce the results with desired resolution. The daily and monthly mean data are available for TCWV-COMBI product during July 2002 - December 2017. Table 1 summarizes the various original data sources that are used in the TCWV-COMBI CDR.

Here we use the daily / 0.05°x0.05° TCWV-COMBI dataset. However, as detailed above the data processing of the TCWV-COMBI is different between land and ocean which then impact the processing of the CMIP6 models: over land areas TCWV is estimated under cloud-free conditions, while over ocean areas TCWV is estimated until heavy precipitation occurs. Moreover





**Table 1.** Summary of the characteristics of CCI water vapour data TCWV-COMBI.

| Data source | Spectral domains | Region | Data description | Spatial resolution | Time span | Reference |
|---|---|---|---|---|---|---|
| MERIS | NIR | Land, coastal and sea-ice | Daytime, cloud free | 1200 m | 2002 - 2012 | Fischer and Bennartz (1997) |
| MODIS | NIR | Land, coastal and sea-ice | Daytime, cloud free | 1000 m | 2011 - 2017 | Gao and Kaufman (2003) |
| OLCI | NIR | Land, coastal and sea-ice | Daytime, cloud free | 1200 m | 2016 - 2017 | Lindstrot et al. (2012) |
| HOAPS | MW | Ocean | 6-hourly composites, without strong precipitation | $0.5°$ degrees | 2002 - 2017 | Lindstrot et al. (2014) |

the evaluation period is restricted to before 2014 for consistency with the available period of the CMIP6 experiment. Such cut in the ESA TCWV-COMBI excludes the OLCI observations.

## 2.2 CMIP6 Models

A subset of seven GCMs participating to CMIP6 is evaluated here, limited by the availability of the required geophysical variables at daily resolution (at least) that is comparable with the CCI_WV CDRs. However, there was no TCWV field at the daily frequency that was available from the Earth System Grid Federation (ESFG) (node of Institut Pierre Simon Laplace, IPSL). Therefore we recomputed TCWV from the vertical profiles of specific humidity q (in g/kg) that were provided at the model vertical resolution. High vertical resolution of specific humidity (more than 19 vertical levels in the troposphere) were

then necessary to be certain to capture the full tropospheric water vapour (using the extraction on a selection of pressure levels would bias the computation of TCWV).

The TCWV (in kg/m$^2$) from each model is thus calculated using:

$$TCWV = \int\limits_{surface}^{top} q \frac{\mathrm{d}p}{g} \tag{1}$$

where g is the gravitational acceleration constant, and dp is the difference between adjacent pressure levels (hPa).

We focus on the AMIP (Atmospheric Model Inter-comparison Project) (Ackerley et al., 2018) scenario with prescribed time-varying sea surface temperature (SST) and sea-ice concentrations from observations and includes variations in natural and anthropogenic external forcings (Eyring et al., 2016). The detailed model descriptions are listed in Table 2. In addition to the CMIP6 models, the ensemble mean of the seven models is also included in the following analysis to represent the mean state of the CMIP6 models.

Since the TCWV-COMBI data is cloud screened over land area, it is important to carefully analyze the cloud conditions for CMIP6 models before making the quantitative comparison. A series of tests have been done (not shown) using different screening thresholds from the cloud fraction and precipitation rate to determine the best cloud conditions for the model-observation comparison. The key aspect here is to have a filter on clouds and/or precipitation that is stringent enough for an apple-to-apple comparison but also that leaves enough point from a statistical point of view. Then, over land the thresholds

are set with maximum cloud fraction of 50% for each vertical level and, over ocean with precipitation rate less than 0.001 kg/m$^2$/s$^2$ (Sohn and Bennartz, 2008). These thresholds result in a reduction of the size of the datasets that differs according to





**Table 2.** Main characteristics of the seven CMIP6 models of the study along with TCWV-COMBI and ERA5. The percentages over land and ocean are computed once the scene selection is applied.

| Institution | Model ID | Horizontal resolution | Vertical resolution | Percentage of land data (%) | Percentage of ocean data (%) | Reference |
|---|---|---|---|---|---|---|
| CCCma | CanESM5 | 2.81° × 2.81° | 49 Levels (1- 1022 hPa) | 55.63% | 99.89% | Swart et al. (2019) |
| CNRM-CERFACS | CNRM-CM6-1 | 1.41°×1.41° | 91 Levels ((0.1 - 1039 hPa)) | 62.85% | 99.86% | Voldoire et al. (2019) |
| | CNRM-ESM2-1 | 1.41°×1.41° | 91 Levels (0.1 - 1039 hPa) | 62.81% | 99.86% | Séférian et al. (2019) |
| IPSL | IPSL-CM6A-LR | 1.25°×2.50° | 79 Levels (0 - 1028 hPa) | 76.10% | 99.79% | Lurton et al. (2020) |
| MPI-M | MPI-ESM1-2-HR | 0.94°×0.94° | 95 Levels (0 - 1055 hPa) | 69.90% | 99.98% | Müller et al. (2018) |
| NCAR | CESM2 | 0.94°×1.25° | 32 Levels (4 - 993 hPa) | 47.14% | 99.97% | Danabasoglu et al. (2020) |
| | CESM2-WACCM | 0.94°×1.25° | 70 Levels (0 - 993 hPa) | 46.14% | 99.97% | Gettelman et al. (2019) |
| ESA CCI_WV | TCWV-COMBI | 0.05°×0.05° | - | 43.73% | 99.82% | Ref as in Table 1 |
| ECMWF | ERA5 | 0.5°×0.5° | - | 52.76% | 97.14% | Eyring et al. (2016) |

the region and the percentages of remaining data for land and for ocean are indicated in Table 2. Over land, the percentage of data remained for the CMIP6 models after the cloud screening are globally in the range 46.14% - 76.10%. Over tropical oceans the percentage of data remained is higher and range from 99.79% to 99.98%. Although the scene selection is more stringent

over land, this indicates that the CMIP6 data used in the following analysis are comparable in terms of size of sample to the data from TCWV-COMBI and ERA5. It is worth mentioning that although the screening thresholds for the models are set to meet the criteria of the TCWV-COMBI product, the number of data retained for the comparison are not exactly the same for all models. Therefore, differences among data sets may be observed in the analysis. This is particularly true over tropical land.

## 2.3 ERA5

The reanalysis data are widely analyzed in atmospheric sciences to assess the impact of changes in observation system, to scale progress in model simulations, and to calculate climatology for forecast-error evaluation (Hersbach et al., 2020). The ECMWF's ERA5 TCWV data are based on the integrated forecasting system (IFS) Cy41r2, with considerable enhanced horizontal resolution of 31 km compared to 80 km for ERA-Interim. Here the ERA5 TCWV with hourly frequency are averaged into daily data. To compare the data under the same conditions, the ERA-5 land-sea mask is employed for land and ocean





separation, and a scene selection is performed and is similar to the process of the CMIP6 data. Hence data with total cloud
cover less than 95% and total column cloud liquid water less than 0.005 kg/m$^2$ over land (Sohn and Bennartz, 2008), and data
with total precipitation less than 0.001 kg/m$^2$/s$^2$ over ocean are retained.

## 3   Methods

The time series of the daily means of the CMIP6, ERA5 and ESA CCI_WV TCWV-COMBI are analyzed with tropical-land
and tropical-ocean separation over the common observation period that covers July 2003 to December 2014.

The intercomparisons are conducted according to two approches:

1) The first approach evaluate the interannual variation of TCWV based on the probability distribution function (PDF) estab-
lished from the daily records for each year of the period. The percentiles of the TCWV are defined from the yearly distributions
and the data is sorted by intervals of 10 percentiles. Finally, the mean TCWV of each interval is computed and normalized

by the corresponding mean TCWV of the whole observation period for this given percentile.This is generalized for every per-
centile. This approach is meant to highlight the tropical anomalies with respect to the mean and trace back to the inter-annual
variability of the tropical atmosphere.

2) The second approach is based on the fact that the water vapour distribution is strongly controlled by the large-scale vertical

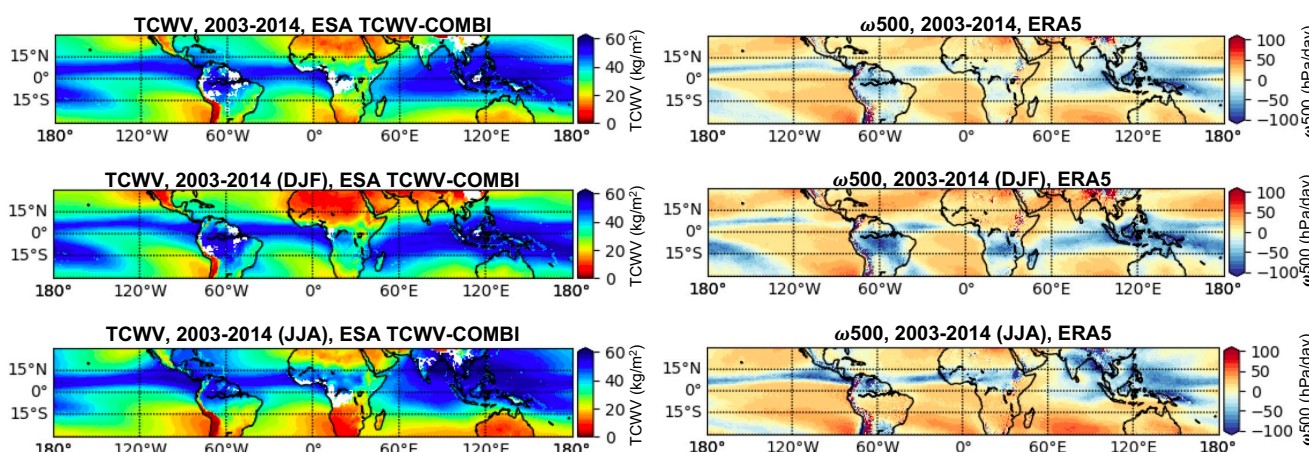

**Figure 1.** Maps of the TCWV-COMBI (in kg/m$^2$) during 2003 - 2014 for the tropical region (30°S - 30°N) for the whole period, Winter
(December, January and February - DJF), and Summer (June, July, and August - JJA) and the corresponding maps of ERA5 $\omega500$ (in
hPa/day).

motion of the atmosphere. Therefore, we can use the mid-tropospheric atmospheric vertical velocity at 500 hPa (noted $\omega500$
in hPa/day) as a proxy for the vertical motions in the tropics (Bony et al., 2004). While such framework has been greatly
used to study tropical clouds and their distribution (e.g., Konsta et al., 2012; Höjgård-Olsen et al., 2020), this link between
vertical motion and TCWV is documented (e.g., Brogniez and Pierrehumbert, 2007) and further illustrated on Figure 1. Figure





1 presents the TCWV-COMBI averaged over the whole 2003-2014 period as well as the mean Winter (December, January, and February - DJF) and mean Summer (June, July, and August - JJA), together with the corresponding $\omega 500$ taken from ERA5 at

a monthly scale. As expected, a moist troposphere is associated with large-scale ascending motion ($\omega 500 < 0$ hPa/day) while a dry troposphere is associated with large-scale subsidence ($\omega 500 > 0$ hPa/day). The TCWV data are sorted upon 10hPa/day-bins of monthly values of $\omega 500$. The dynamical decomposition is performed for all TCWV data records at each year of the time period. Moreover the TCWV data averaged over the whole 2003-2014 period is also sorted into the corresponding $\omega 500$ bins of the period and this value is considered as the reference to normalize the results. This second approach allows to study

the trends in TCWV for a given state of the large-scale dynamics, and thus overcome issues associated with variations (such as shifts or expansion) of the atmospheric circulation (Vallis et al., 2015; Mbengue and Schneider, 2017).

## 4    Results and Discussions

This section aims to assess the degree of agreement in the TCWV climatology and interannnual variations between the ESA CCI_WV TCWV-COMBI, CMIP6 models and ERA5 reanalysis data over the tropics (30°S - 30°N). The distribution of the

water vapour over tropics and its link to large-scale circulation ($\omega 500$) are discussed in detail.

### 4.1    Description of the tropical TCWV : 2003-2014

#### 4.1.1    Time series

Figure 2 shows the climatology of the TCWV of the different datasets over land (Fig2a; clear skies only) and over ocean (Fig2b; all-weather without heavy precipitation) for the period 07/2003-12/2014. Overall, the different data records agree well

with each other despite some discrepancies. Strong seasonal variations are observed over tropical land with minima reached during JFM and maxima reached during JJA and with a very weak interannual variability. Over the tropical oceans, the seasonal variations are softer (the minima still occur during JFM, but the maxima are not always reached in JJA) with a strong interannual signal. More specifically, the ESA TCWV-COMBI data is globally moister compared to the CMIP6 models and to ERA5 data, and this moist bias is even more pronounced over tropical land (Fig 2a, $\sim 10$kg/m$^2$ over land vs $\sim 2$kg/m$^2$ over ocean). On

the other hand, the observed ESA TCWV-COMBI over ocean reaches globally higher values than over land. This difference can be explained because the ESA TCWV-COMBI dataset is composed with clear-sky-only data, which are likely drier than the nearby cloud area for a given location and thus translates into a dry bias associated to moistening processes by convective clouds (Sohn et al., 2006). Besides, the boundary layer is drier in the continental subtropics while the maritime stratocumulus zones are wetter at low levels and very dry at high altitude.

#### 4.1.2    The distribution of tropical TCWV

The normalized PDFs of the daily TCWV obtained from all data records over both land and ocean are displayed in Figure 3. Similar to the time series analysis, the characteristics over land are significantly different with the results over ocean area





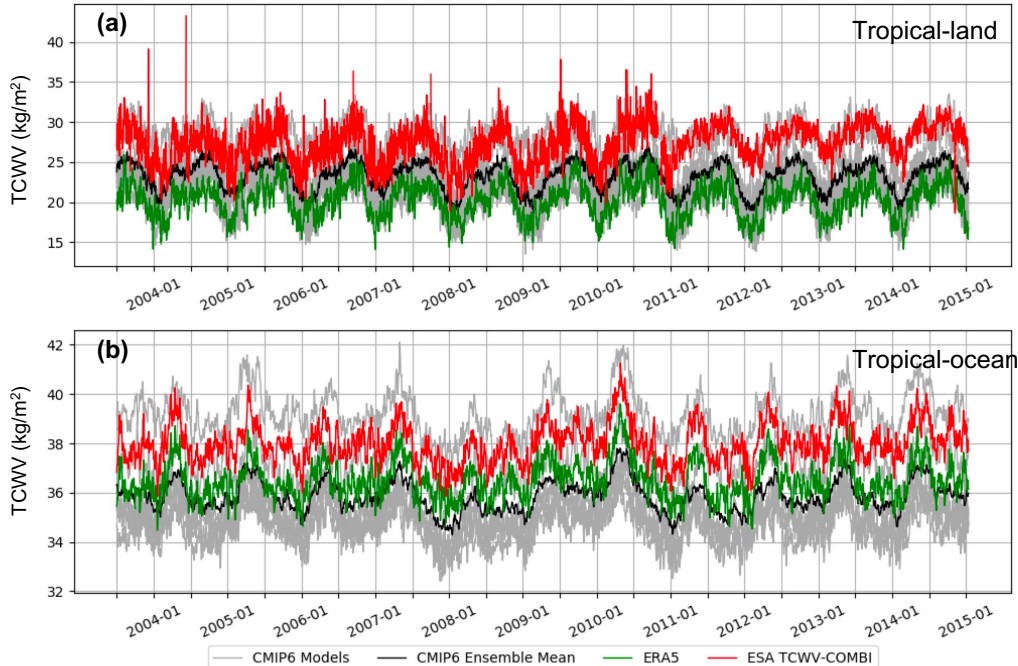

**Figure 2.** Time series of daily mean TCWV in the tropics (30°S - 30°N) over (a) land areas under clear-sky condition and (b) ocean areas except for heavy precipitation (see details in Section 2). The time series covers the period 07/2003-12/2014. The gray lines denote the individual CMIP6 models while the black line represents their ensemble mean. The green line represents ERA5 and the red line is the ESA CCI_WV TCWV-COMBI.

because of the data screening. Over land, all datasets reach a first maximum at around 10-13 kg/m$^2$ and present a secondary maximum near 40-50 kg/m$^2$. Moreover, more lower values are observed in the CMIP6 models and ERA5 than in the ESA TCWV-COMBI dataset and this cannot be explained by the cloud-screening method alone. Indeed, the ESA TCWV-COMBI dataset is strictly cloud-free while some cloudy scenes remain in both ERA5 and CMIP6 (see Section 2), which could translate in a moister TCWV of the latter data records.

Over oceans, most of the TCWV data are located around 20-60 kg/m$^2$. The main peak is around 30 kg/m$^2$, and a secondary peak appears near 50 kg/m$^2$. While the frequency of situations of the main peak is nearly identical for TCWV-COMBI, ERA5 and CMIP6, there is a divergence for the secondary peak. This secondary peak even dominates the PDF of the CMIP6 models while ERA5 and ESA TCWV-COMBI are still quite similar. The bimodal distributions can be explained by the presence of more humid columns in the Intertropical Convergence Zone (ITCZ) and relatively drier ones in subtropical regions.

### 4.1.3 Extremes of the distributions

The data records are then evaluated following the approach (1) described in Section 3: the percentiles of the annual distributions of TCWV (at daily resolutions) are sorted into bins of 10% intervals, and this is done for each year of the period.





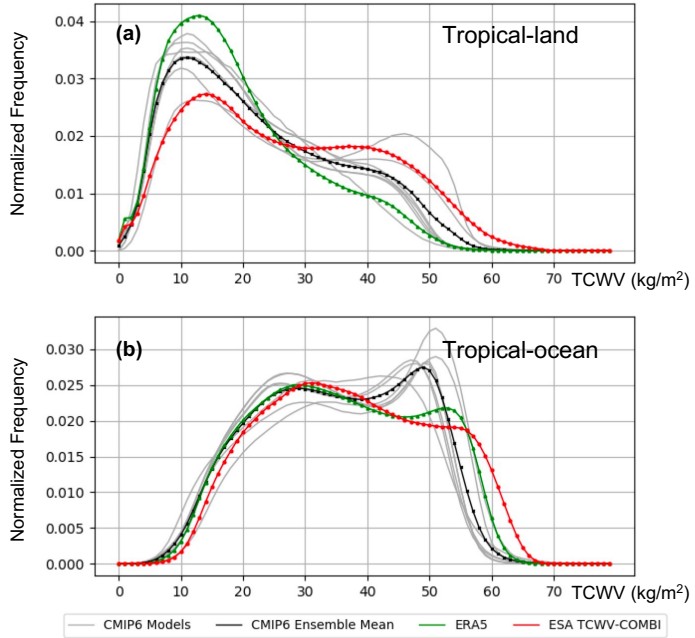

**Figure 3.** The normalized PDFs of the TCWV in the tropical area (30°S-30°N) over (a) land areas (under clear-sky only condition), and over (b) ocean areas (under all-weather condition except for heavy precipitation). The gray lines denote the individual CMIP6 models while the black line represents their ensemble mean. The green line represents ERA5 and the red line is the ESA CCI+ TCWV-COMBI.

The normalized TCWV for land areas and sorted by 10%-percentiles intervals for each year are displayed in Figure 4. The bluish colors indicate that the TCWV value of the interval is larger than the reference value, indicating wet anomalies. The redish colors indicate that the TCWV value is smaller than the reference, indicating dry amonalies. As shown in the figure, the ESA TCWV-COMBI data have quite different characteristics compared to CMIP6 and ERA5 results. A clear moistening

trend is observed in the drier percentiles of TCWV-COMBI record. The tiping point seems to be 2011 and thus may be caused by the inhomogeneity of cloud-mask products for different observation instruments when computing the ESA TCWV-COMBI data. Indeed, the data record merged NIR observation from MERIS over 2002-2012 and MODIS observation are included over 2011-2017. Despite individual discrepancies, the CMIP6 ensemble mean is in good agreement with the ERA5 data. Overall, anomalies are observed in the time period for all data records: 2008 appears as a dry year over all the range of TCWV while

2010 reveals a clear signal of humidification over the high parts of the distributions of TCWV (percentiles > 60%) of TCWV.

   A similar comparison is performed over the tropical oceans and the results are shown in Figure 5. The TCWV-COMBI results are well coincident with CMIP6 models and ERA5 data. Dry anomalies are observed in 2004 for the ESA TCWV-COMBI and ERA5 in the dry end of TCWV. Dry anomalies are also observed in 2008 and 2011 over the highest percentiles (> 60%) for all data records, while 2008 is the driest year of the period. Wet anomalies are observed in 2010 for the highest percentiles in

all data records. ERA5 also reveals 2012 as a moister year, in the low range of TCWV, but this anomalous year is not present

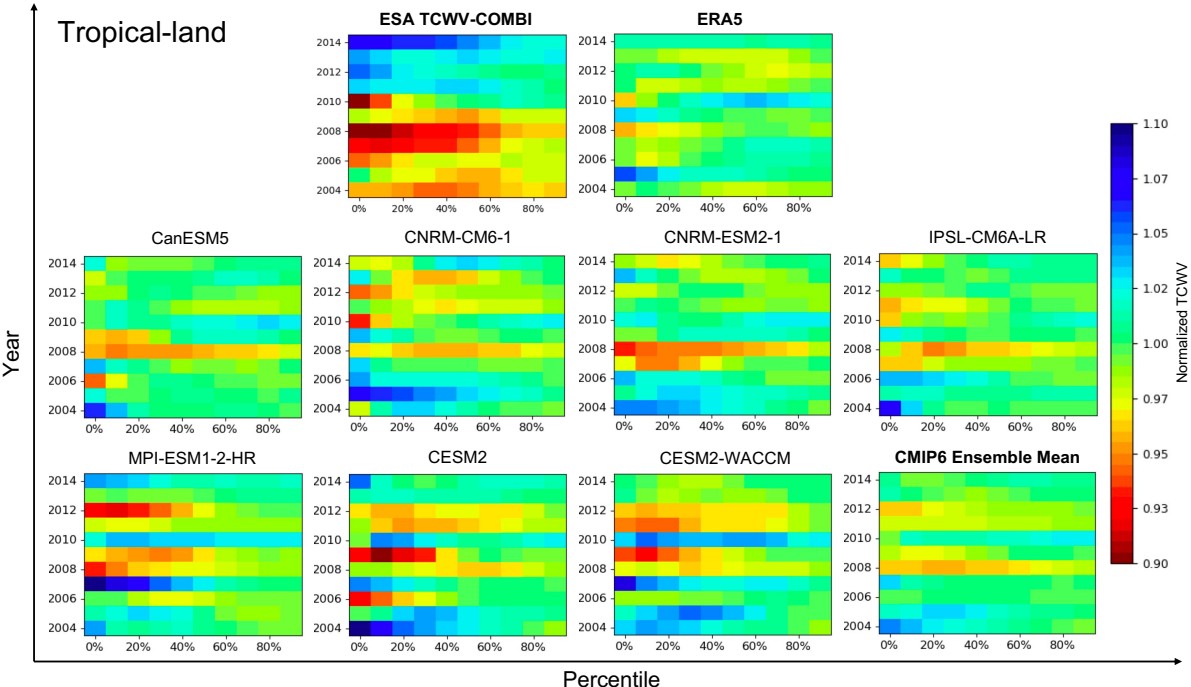

**Figure 4.** Normalized percentiles of the TCWV over land areas for every data record. The percentiles are grouped into bins of 10% intervals. The x-axis represents the percentiles intervals, and the y-axis represents the year. Note that the period starts in 2004 instead of 2003 to focus on full years.

in the ESA TCWV-COMBI or CMIP6 ensemble mean. The very good agreement among the various data sets is largely due to the fact that the CMIP6 models that are evaluated under the AMIP scenario, so with the same prescribed SST for all models and ERA5, and that the relationship between SST and TCWV is largely explained by the Clausius-Clapeyron law (Stephens, 1990). Hence this explains that anomalous years are the same, most notably those concerned by El Nino Southern Oscillation
(Trenberth et al., 2005): 2008 is characterized by a very strong episode of negative ENSO index, as well as 2011, while 2010 is an intermediate year, with the end of a positive ENSO episode followed by a negative one.

## 4.2 TCWV and large-scale circulation

### 4.2.1 General assessment

The interannual variability of TCWV is then analyzed from its links with the large-scale atmospheric circulation, and follows
the approach (2) described in Section 3. The monthly $\omega500$ of individual data records are decomposed into 10 hPa/day intervals in the range of -120 to 120 hPa/day. Figure 6 displays the normalized PDFs of the $\omega500$ of the CMIP6 models and ERA5. As mentioned earlier there is no atmospheric circulation data from the ESA TCWV-COMBI data record, so the $\omega500$ from ERA5

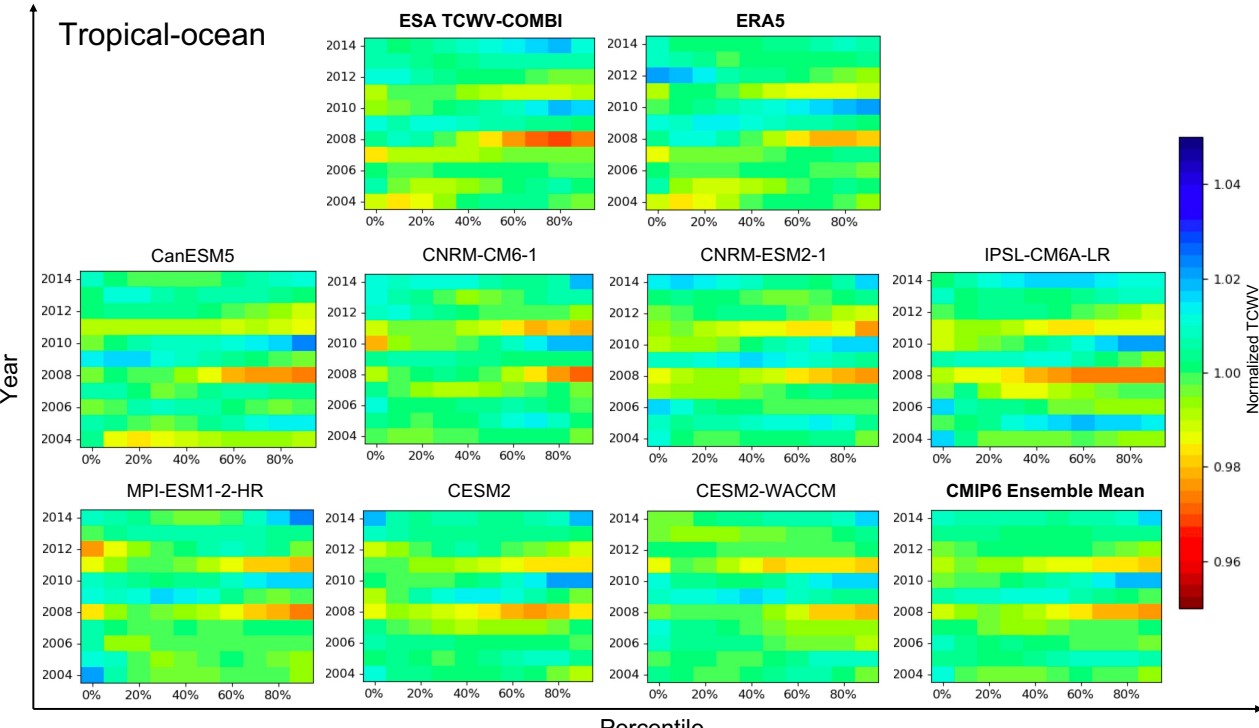

**Figure 5.** Same as Figure 4 but for oceans.

is also employed as the reference for this dataset. Figure 6 (a) and (b) show that the PDFs of the $\omega500$ from the CMIP6 ensemble mean agrees well with the ERA5 data. Most of the $\omega500$ reside in around 10 - 20 hPa/day over both tropical-land and

tropical-ocean area, which characterizes the dominance of the large-scale Hadley subsidence in subtropical free troposphere and explains the clear-sky radiative cooling of the tropics as discussed in Bony et al. (2004).

The TCWV of each dataset is then sorted into the vertical velocity bins by the corresponding value of $\omega500$. Therefore, the variations of the TCWV can be analyzed from the perspective of atmospheric vertical motion. As shown in Figure 6 (c) and (d), large-scale downward motion is associated with a dry troposphere, while large-scale ascent is globally associated with a moister

troposphere. The results are in agreement with the maps of Figure 1. Overall, the CMIP6 models and ERA5 show differences with the ESA TCWV-COMBI data in the amplitude of the signal and in the gradient of moisture between the ascending and descending regions. While the large-scale atmospheric dynamics are consistent among the datasets, the discrepancies in the TCWV reveal difficulties in the moistening processes of the tropical atmosphere : lateral mixing (Pierrehumbert and Roca, 1998; Pierrehumbert, 1998), outflows from clouds, too high/too low precipitation efficiencies of the convective schemes

(Brogniez and Pierrehumbert, 2007). The results suggest that large-scale advection humidification / drying processes are not correctly represented. Besides, the scene selection also explains part of the differences between ESA TCWV-COMBI, ERA5 and CMIP6 models.

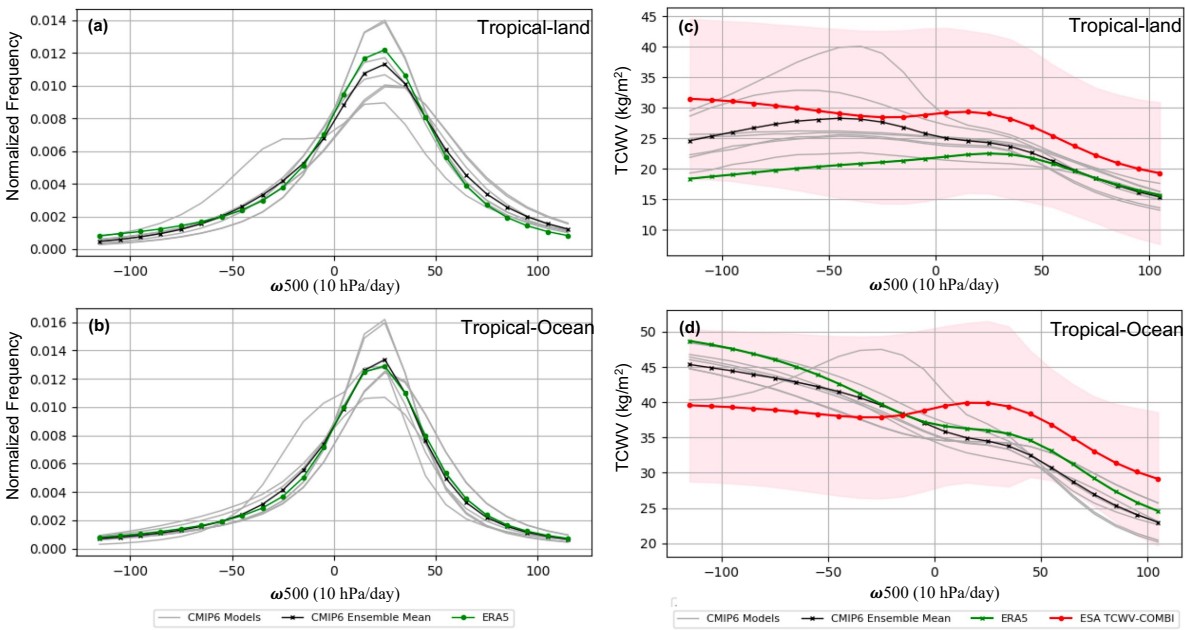

**Figure 6.** (Left) Normalized PDFs of $\omega500$ (in hPa/day) over land (a) and ocean (b) for CMIP6 models (grey lines), their ensemble mean (black line), as well as ERA5 (green line); (Right) Mean TCWV from the CMIP6 models (grey lines), their ensemble mean (black line), ERA5 (green line), and ESA TCWV-COMBI (red line) in different circulation regimes of $\omega500$ over land (c) and ocean (d) areas. The shaded area in pink represents the $\sigma$ of each bin in TCWV-COMBI data.

### 4.2.2  Trends over lands

This global assessment is further discussed by applying the TCWV-$\omega500$ approach for every year of each data record to
delineate the trends in TCWV. As shown in Figure 7, all the data records (except for ERA5) agree that the most positive $\omega500$ bins (meaning the areas of highest downward motion) are associated with the driest troposphere (red) while the most negative $\omega500$ bins (meaning the areas of highest upward motion) are associated with the moistest troposphere (blue).

However the maxima and minima are different : ERA5 and the CMIP6 ensemble mean display the lowest TCWV values ($<$ 16kg/m$^2$) that occur all along the 2004-2014 period while the ESA TCWV-COMBI data record reaches values $\sim$18kg/m$^2$ and
this minimum is reached over 2007-2008. There is also a very strong variability amongst the CMIP6 models : IPSL-CM6A-LR is clearly the moistest model and CanESM5 is the driest. The moist bias of IPSL-CM6A-LR is already documented (Boucher et al., 2020) and is explained by the (too) efficient parametrization scheme of the transport of evaporated air from the surface to the top of the boundary layer. The CanESM5 behavior is opposite and this shall be partly explained by its strong effective climate efficiency (Virgin et al., 2021) which yields to adjust too strongly the atmospheric response to a perturbation. ERA5
displays also a very dry troposphere whatever the dynamical regime. Moreover, all models show that the transition dry/moist occurs around 60hPa/day and this transition is similar also for ERA5 and ESA TCWV-COMBI.





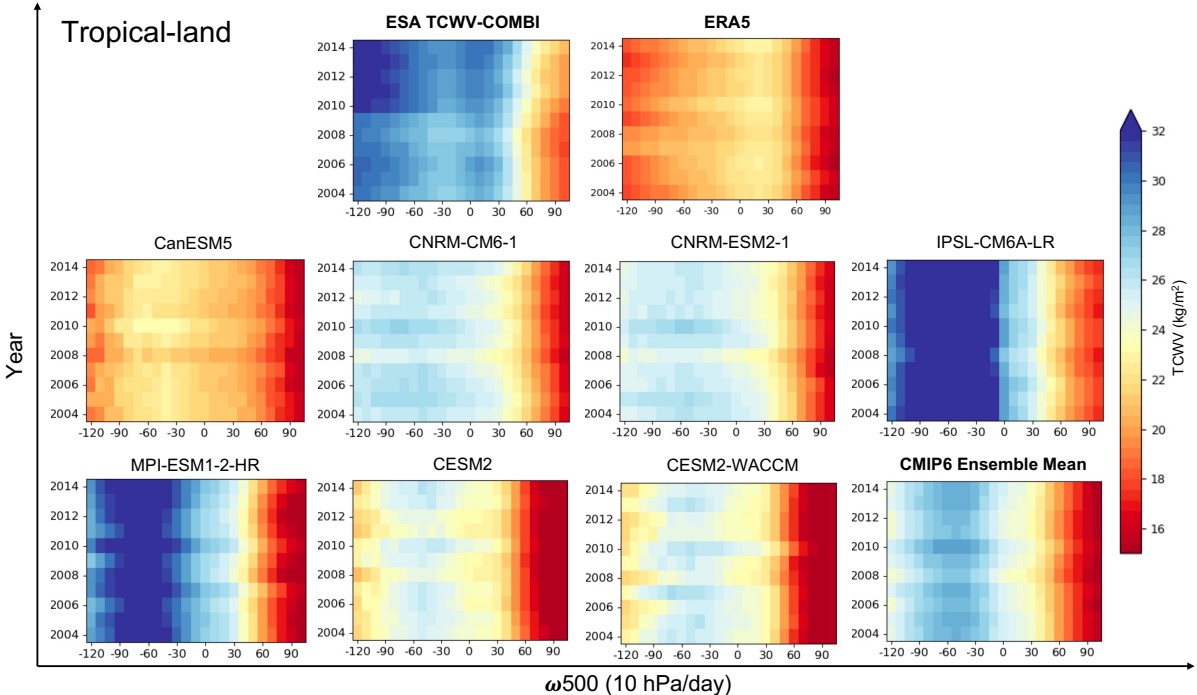

**Figure 7.** Mean of TCWV over tropical land areas at each dynamical intervals ($\omega500$) in 10 hPa/day computed from each data record.

To unravel the anomalies of TCWV, the mean values of TCWV of each circulation regimes observed during the whole comparison period (2004 - 2014) are employed as the reference to normalize the results. The normalized TCWV at each circulation intervals for the different data records are displayed in Figure 8. Although different patterns are observed, dry anomalies occur in 2008 and wet anomalies occur in 2010 for all of the data records. These extreme years are consistent with the previous findings (section 4.1.3) and this second approach provides another angle of analysis on the assessment:

- The ESA TCWV-COMBI reveals a clear moistening tendency since 2004 and this tendency mostly occurs over the subsiding branch of the atmospheric circulation, with a stronger trend after 2011. Although the data record itself, merging several instruments over the period (MODIS is included in 2011), may induce a bias (from sampling for example), a step-like bias would appear and not a tendency.

- The TCWV from ERA5 shows only extreme years but no distinct tendency. In the regions of highest upward motions, 2007 and 2008 appear moister than the other years while 2004 and 2013 appear drier. Once again, this may be due to the scene selection applied over land, but this would not explain entirely the differences with the ESA TCWV-COMBI.

- Finally, the CMIP6 models and their ensemble mean show consistent interannual variabilities : 2008 is a dry anomaly, more or less for all dynamical regimes, and 2010 is a moist anomaly for all models which is in a relative agreement with ERA5 and ESA TCWV-COMBI.





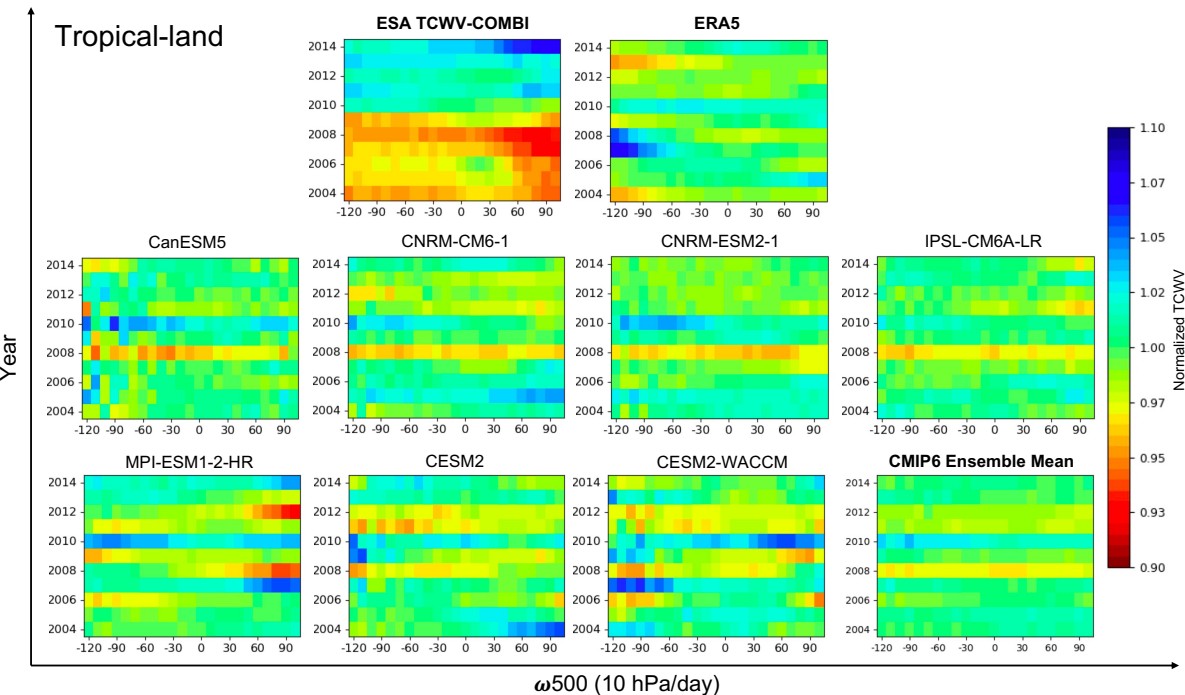

**Figure 8.** Normalized TCWV with respect to the 2004-2014 mean over tropical land areas at each dynamical intervals ($\omega$500) in 10 hPa/day computed from each data record.

### 4.2.3 Trends over oceans

Oceanic situations as considered similarly with respect to the large-scale circulation. The results are shown in Figure 9 (mean TCWV) and Figure 10 (normalized TCWV). As the data over ocean areas are obtained under all-weather conditions except
for heavy precipitation, the impacts from clear-sky biases are greatly reduced. The strongest ascending zones corresponding to the very humid regions are not retained by the TCWV-COMBI data. Different from the results over land where the CMIP6 models showed strong differences (CanESM5 was driest and IPSL-CM6A-LR was the moistest), the amplitudes and gradients of moisture are closer to each other over oceans. Boucher et al. (2020) have underlined, for the IPSL-CM6A-LR model, the role of the transport model in the boundary layer that affects both shallow and deep convective regimes. In the present study,
this could result in compensations of biases, a moist bias being still present in the regimes of slightly upward motion ($\sim$ -30 hPa/day).

The normalized temporal tendencies are shown in Figure 10. The temporal evolutions of TCWV-$\omega$500 are consistent with the earlier analysis based on the temporal evolution of the percentiles of TCWV over ocean (section 4.1.3). The extreme dry and moist years (respectively 2008 and 2010) are the same between ESA TCWV-COMBI, ERA5 and the CMIP6 ensemble mean.





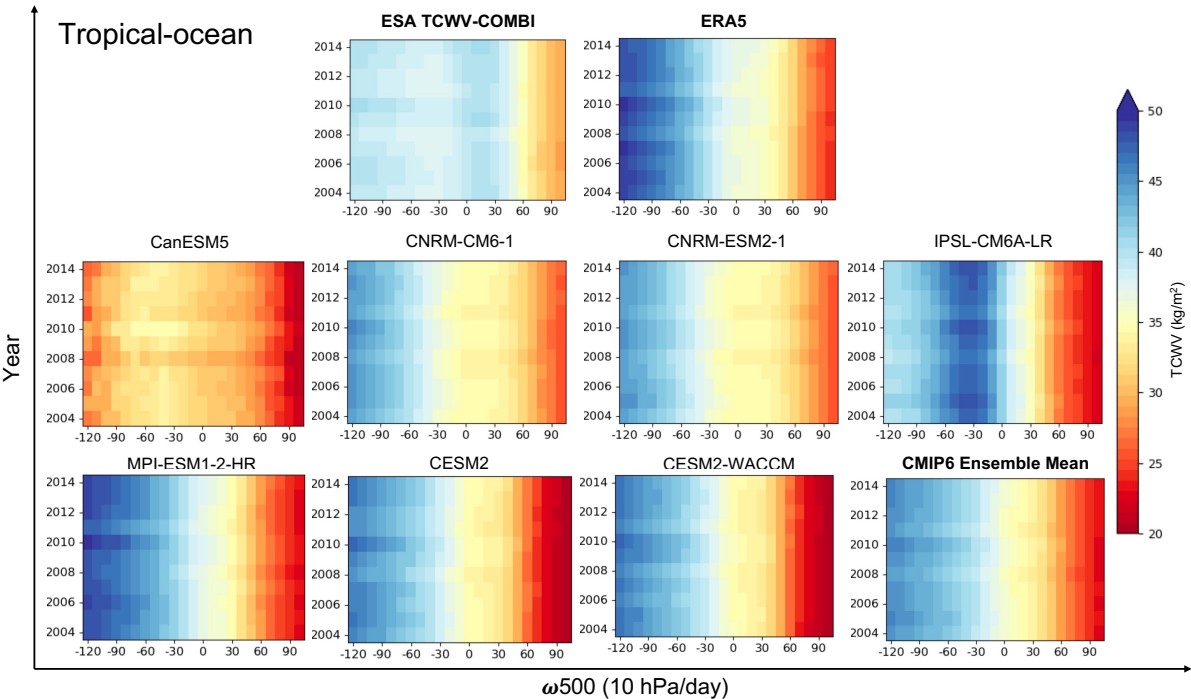

**Figure 9.** Mean of TCWV over tropical ocean areas at each dynamical intervals ($\omega 500$) in 10 hPa/day computed from each data record.

## 5 Conclusions

Despite the importance of water vapour in the study of climate variability, our ability to evaluate the water vapour feedback is constrained by its measurements at ranges of scales that are adapted for local, regional and global studies. This deficiency is attributable in part to the fact that it is difficult to quantitatively and accurately measure the distribution of water vapour. To work towards the requirement of GCOS on satellite-based water vapour observation as ECV, the ESA Climate Change Initiative "Water Vapour" project (ESA CCI_WV) tackled this challenge by generating gridded products on stratospheric and tropospheric water vapour from multiple satellite observations suitable to climate and process studies.

We have conducted a comprehensive evaluation of the tropical water vapour (30°N-30°S) of seven GCMs (CMIP6 models, AMIP scenario) and ERA5 using the TCWV-COMBI climate data record developed within the ESA CCI_WV project as a reference. The study focused over tropical-land and tropical-ocean areas at the daily frequency and over the 2003-2014 period.

The variability of TCWV was analyzed according to (i) its probability density function (PDF) defined at a yearly scale over the period and (ii) to the large-scale circulation using the atmospheric vertical velocity at 500hPa ($\omega 500$) as a proxy of the tropospheric overturning circulation.

Different patterns of variability are observed among the various datasets, the largest discrepancies being noticed over land areas, while over the oceans the datasets are closer to each other:





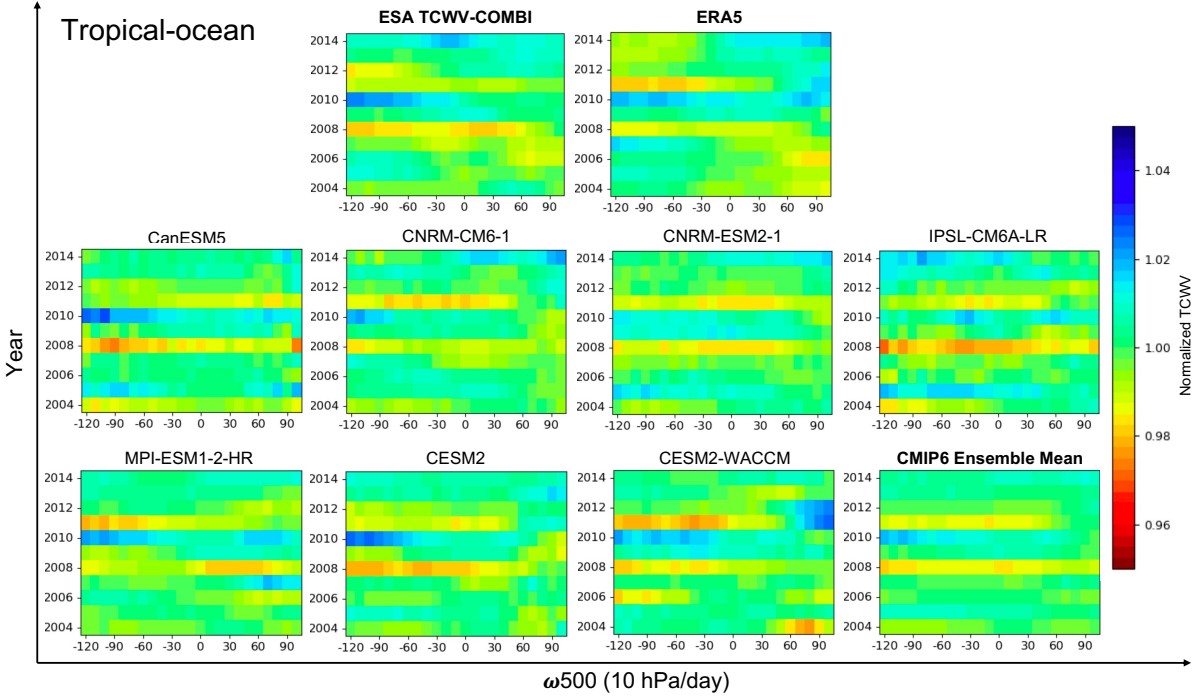

**Figure 10.** Normalized TCWV with respect to the 2004-2014 mean over tropical ocean areas at each dynamical intervals ($\omega500$) in 10 hPa/day computed from each data record.

- over land, the PDFs of the ESA TCWV-COMBI present a clear moistening trend of their driest percentiles, with a tiping point in 2011, probably associated to the addition of the MODIS observation in the climate data record. The projection of the TCWV onto regimes of $\omega500$ shows the same behavior, the drying trend being present for all regimes of $\omega500$. Interestingly, the CMIP6 ensemble mean and the ERA5 reanalysis are in good agreement in terms of interannual anomalies, although the ERA5 TCWV is clearly too dry.

- over ocean, the PDFs of all datasets present the same interannual variability. The extreme dry and moist years, associated to El Nino and La Nina events, are the same. This similarities hold when using the large-scale circulation as an evaluation tool, with the same transition between the dry/subsiding regimes and the moister/ascending regimes.

The results show that the ESA TCWV-COMBI data, ERA5 data vary within the ensemble spread of CMIP6 models, indicating that the mean models could correctly represent the evolution of water vapour with respect to large-scale circulation. The humid area is related with the ascending motion (negative value in $\omega500$) and dry area is related with the subsiding motion (positive value in $\omega500$) over both tropical land and tropical ocean area. There are discrepancies observed among the data records, because of the lateral mixing, outflows from clouds, and the precipitation efficiencies of the convective schemes. It is difficult to track entirely the reasons of the differences, however, the differences and similarities can be explained by several factors :

1) the use of different satellites with different accuracies and resolutions within the ESA TCWV-COMBI may explain part of

the moistening trend observed for this dataset over land.

2) the cloud-masks applied to the GCMs and ERA5 and defined to mimic the cloud-mask of the observation can also explain the differencies.

3) the parametrization of the moisture fluxes at the surface and of convection, as well as the climate efficiency of the GCMs also contribute to the observed differences.

4) the use of AMIP scenarios, defined from prescribed sea surface temperatures, as well as a scene selection that is much more conservative than over land, explain almost entirely the very good agreement between the ESA TCWV-COMBI, ERA5 and the GCMs.

It is really necessary to underline the role of the cloud mask in the assessment of water vapour fields in climate models using observations, even though water vapour seems to be an easier parameter than clouds. Climate models provide water

vapour profiles (and sometimes the integrated values) at the scale of their mesh which is usually a lot larger (see Table 2) than the observed water vapour, and whatever the cloud distribution within the mesh. However, the water vapour estimated from observations by satellite sensors is strictly analyzed with respect to cloud contamination. This clearly poses the question about having access to the simulated water vapour (full profiles as well as integrated values) for the clear sky part of the meshes of the climate models.

*Author contributions.* JH carried out the data analysis and prepared all the figures. JH, HB and LP contributed to the interpretation of the results and wrote the paper.

*Competing interests.* The authors do not declare any competing interests

*Acknowledgements.* The study was funded by ESA via the Water_Vapour_CCI project of ESA's Climate Change Initiative (CCI). The combined microwave and near-infrared imager based product COMBI was initiated, funded and ESA Water_Vapour_CCI project, with

contributions from Brockmann Consult, Spectral Earth, Deutscher Wetterdienst and the EUMETSAT Satellite Climate Facility on Climate Monitoring (CM SAF). The combined MW and NIR product will be owned by EUMETSAT CM SAF and will be released by CM SAF in late 2021. This study benefited from the ESPRI (Ensemble de Services Pour la Recherche à l'IPSL) computing and data center (https: //mesocentre.ipsl.fr).





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
