# Peer review of "Evaluation of tropical water vapour from CMIP6 GCMs using the ESA CCI "Water Vapour" climate data records"

_Atmospheric Chemistry and Physics, 2021_

## Author Comment (AC1)

**Author's Response**

**Paper Number: ACP-2021-976 Paper Title: Evaluation of tropical water vapour from CMIP6 GCMs using the ESA CCI "Water Vapour" climate data records**

We thank the editor and the anonymous referees for their invaluable comments on our manuscript that help to improve the quality of this manuscript. We have carefully addressed all the points raised by the academic editor and reviewers. Please find below our response and corresponding changes made in the revised paper.

Sincerely, On behalf of all authors, Jia He

**Anonymous Referee #2**

This paper presents a comparison of the total column water vapour from the ESA TCWV- COMBI data set to ERA5 and a suite of CMIP6 models. TCWV is analyzed in terms of PDFs' percentiles and composites on large-scale circulation. The study shows that applying a consistent cloud screening is important for comparing the different data sets. Over oceans, both ERA5 and CMIP6 models agree well with the observations, regarding variability and relation to circulation. Over land, larger differences and discrepancies occur, likely related to changes in instruments.

I think this is an interesting comparison paper of a new observational data set with new model simulations and reanalysis, and suitable for publication in ACP. I have no major concerns, only a few minor and specific comments below, which hopefully will help to improve the paper.

Minor comments:

1. English language: I had the impression that the language and wording could still be improved. I've just listed a few typo/wording issues below which I came across, but would recommend a careful final check.

Reply: We thank the reviewer for the comment. We have carefully checked the manuscript for language and wording. We hope that this is now improved.

2. Cloud screening: I understand from the description in L110 that the cloud screening of the model data (for both CMIP6 and ERA5) has been adjusted to optimize the comparison to TCWV-COMBI, with the additional constraint to have enough data included - and that this screening is critical for the comparison. I find this description somewhat qualitative and insufficient. I'd recommend to describe the screening approach in a clearer way (perhaps even including a figure for illustration), to enable readers to repeat the analysis.

Reply: We understand the reviewer's comment and we agree that more details should be provided on this point of methodology since it is an important aspect of the model-observation comparison.

As the TCWV-COMBI data provide observation under clear-sky only conditions over the land area, we have computed that there are only 43.73% of valid data over land that can be used to make the comparison. It's necessary to conduct the cloud screening process for CMIP6 models and ERA5 data over the land area to make them comparable with the satellite-based TCWV-COMBI product. Therefore, the total cloud cover (tcc), and total column cloud liquid water (tclw) are adopted as indicators for cloud condition for ERA5 data, while cloud fraction (cf) is adopted for CMIP6 models. Following the study of Sohn and Bennartz (2008) (Sohn, Byung-Ju, and Ralf Bennartz. "Contribution of water vapour to observational estimates of longwave cloud radiative forcing." Journal of Geophysical Research: Atmospheres 113.D20 (2008).), pixels with tcc < 95% and tclw < 0.005 kg/m2 over land are retained for the cloud screening of ERA5 data. The hourly tcc and tclw data were firstly averaged into daily

value, and then the clouded pixels of ERA5 data were considered at the daily scale as contaminated data and were removed in the following analysis.

Since we don't have the same variables of ERA5 available for CMIP6 models, a series of threshold tests were conducted with different cloud fraction (cf). The percentage of valid data for a given cloud fraction applied to the entire atmospheric column are summarized in the Table below, for the 7 CMIP6 models. The associated Figures below present a selection of maps for the CanESM5 model and the 4 thresholds of the Table, together with the corresponding ERA5 and TCWV-COMBI maps.

| Model ID      | Clear-sky data remained over tropical land
for the evaluation analysis |        |        |        |
|---------------|---------------------------------------------------------------------------|--------|--------|--------|
|               | cf

Figure. Examples of daily mean TCWV over tropical land obtained from CanESM5 with different cloud mask (pixels with cloud fraction larger than 75%, 50%, 25%, and 5% at all pressure levels are considered as clouded) along with data from TCWV-COMBI and ERA5, July 1st 2003.

It is important to mention that our priority is to remove cloudy pixels, with due consideration to maintaining enough data for further evaluation. As shown in the above table and figure, it's judicious to consider the pixel as a clear sky when the cf is less than 50% at all pressure levels for CMIP6 models: this threshold ensures a reasonable spatial coverage and allows to keep areas that are visible in the TCWV-COMBI and ERA5 like in Central Africa or South America.

The following discussion has been added for clarification purposes:

L111: The cloud-screening step of the model-to-observation approach requires an important trade-off because cloudy pixels must be removed cloudy pixels but with due consideration to maintaining enough data for further evaluation. As there is no tcc and tclw product available for CMIP6 models, the cloud

fraction (cf) is employed instead as an indicator of the cloudiness. We conducted a series of threshold tests, by screening pixels with cf values larger than 5%, 25%, 50%, and 75% at all pressure levels. The distribution of clear-sky data over tropical land from CanESM5 of the CMIP6 model with different cloud masks along with TCWV-COMBI and ERA5 are shown in Figure 1. As shown in the figure, the data with cf less than 50% at all pressure levels could ensure a reasonable spatial coverage comparable to TCWV-COMBI and ERA5. Therefore, we adopted this threshold as the general cloud mask for the CMIP6 models, although this cannot be considered a purely clear sky.

---

## Author Comment (AC2)

**Author's Response**

**Paper Number: ACP-2021-976**
**Paper Title: Evaluation of tropical water vapour from CMIP6 GCMs using the ESA CCI "Water Vapour" climate data records**

We thank the editor and the anonymous referees for their invaluable comments on our manuscript that help to improve the quality of this manuscript. We have carefully addressed all the points raised by the academic editor and reviewers. Please find below our response and corresponding changes made in the revised paper.

Sincerely,
On behalf of all authors,
Jia He

**Anonymous Referee #2**
This paper presents a comparison of the total column water vapour from the ESA TCWV- COMBI data set to ERA5 and a suite of CMIP6 models. TCWV is analyzed in terms of PDFs' percentiles and composites on large-scale circulation. The study shows that applying a consistent cloud screening is important for comparing the different data sets. Over oceans, both ERA5 and CMIP6 models agree well with the observations, regarding variability and relation to circulation. Over land, larger differences and discrepancies occur, likely related to changes in instruments.
I think this is an interesting comparison paper of a new observational data set with new model simulations and reanalysis, and suitable for publication in ACP. I have no major concerns, only a few minor and specific comments below, which hopefully will help to improve the paper.

Minor comments:

1. English language: I had the impression that the language and wording could still be improved. I've just listed a few typo/wording issues below which I came across, but would recommend a careful final check.

Reply: We thank the reviewer for the comment. We have carefully checked the manuscript for language and wording. We hope that this is now improved.

2. Cloud screening: I understand from the description in L110 that the cloud screening of the model data (for both CMIP6 and ERA5) has been adjusted to optimize the comparison to TCWV-COMBI, with the additional constraint to have enough data included - and that this screening is critical for the comparison. I find this description somewhat qualitative and insufficient. I'd recommend to describe the screening approach in a clearer way (perhaps even including a figure for illustration), to enable readers to repeat the analysis.

Reply: We understand the reviewer's comment and we agree that more details should be provided on this point of methodology since it is an important aspect of the model-observation comparison.
As the TCWV-COMBI data provide observation under clear-sky only conditions over the land area, we have computed that there are only 43.73% of valid data over land that can be used to make the comparison. It's necessary to conduct the cloud screening process for CMIP6 models and ERA5 data over the land area to make them comparable with the satellite-based TCWV-COMBI product. Therefore, the total cloud cover (tcc), and total column cloud liquid water (tclw) are adopted as indicators for cloud condition for ERA5 data, while cloud fraction (cf) is adopted for CMIP6 models. Following the study of Sohn and Bennartz (2008) (Sohn, Byung-Ju, and Ralf Bennartz. "Contribution of water vapour to observational estimates of longwave cloud radiative forcing." Journal of Geophysical Research: Atmospheres 113.D20 (2008).), pixels with tcc < 95% and tclw < 0.005 $kg/m^2$ over land are retained for the cloud screening of ERA5 data. The hourly tcc and tclw data were firstly averaged into daily

value, and then the clouded pixels of ERA5 data were considered at the daily scale as contaminated data and were removed in the following analysis.

Since we don't have the same variables of ERA5 available for CMIP6 models, a series of threshold tests were conducted with different cloud fraction (cf). The percentage of valid data for a given cloud fraction applied to the entire atmospheric column are summarized in the Table below, for the 7 CMIP6 models. The associated Figures below present a selection of maps for the CanESM5 model and the 4 thresholds of the Table, together with the corresponding ERA5 and TCWV-COMBI maps.

Table: Summary of valid data remained after applying different cloud masks for CMIP6 models over tropical land.

| Model ID | Clear-sky data remained over tropical land for the evaluation analysis | | | |
|---|---|---|---|---|
| | cf<75% | cf<50% | cf<25% | cf<5% |
| CanESM5 | 73,83% | 55,63% | 36,43% | 28,73% |
| CNRM-CM6-1 | 86,24% | 62,85% | 36,82% | 18,43% |
| CNRM-ESM2-1 | 85,97% | 62,81% | 37,18% | 18,71% |
| IPSL-CM6A-LR | 92,18% | 76,10% | 49,03% | 24,54% |
| MPI-ESM1-2-HR | 89,77% | 69,90% | 43,69% | 22,78% |
| CESM2 | 63,04% | 47,14% | 31,48% | 19,90% |
| CESM2-WACCM | 61,76% | 46,14% | 30,88% | 19,56% |

[Figure]

Figure. Examples of daily mean TCWV over tropical land obtained from CanESM5 with different cloud mask (pixels with cloud fraction larger than 75%, 50%, 25%, and 5% at all pressure levels are considered as clouded) along with data from TCWV-COMBI and ERA5, July 1st 2003.

It is important to mention that our priority is to remove cloudy pixels, with due consideration to maintaining enough data for further evaluation. As shown in the above table and figure, it's judicious to consider the pixel as a clear sky when the cf is less than 50% at all pressure levels for CMIP6 models: this threshold ensures a reasonable spatial coverage and allows to keep areas that are visible in the TCWV-COMBI and ERA5 like in Central Africa or South America.

The following discussion has been added for clarification purposes:

L111: The cloud-screening step of the model-to-observation approach requires an important trade-off because cloudy pixels must be removed cloudy pixels but with due consideration to maintaining enough data for further evaluation. As there is no tcc and tclw product available for CMIP6 models, the cloud

fraction (cf) is employed instead as an indicator of the cloudiness. We conducted a series of threshold tests, by screening pixels with cf values larger than 5%, 25%, 50%, and 75% at all pressure levels. The distribution of clear-sky data over tropical land from CanESM5 of the CMIP6 model with different cloud masks along with TCWV-COMBI and ERA5 are shown in Figure 1. As shown in the figure, the data with cf less than 50% at all pressure levels could ensure a reasonable spatial coverage comparable to TCWV-COMBI and ERA5. Therefore, we adopted this threshold as the general cloud mask for the CMIP6 models, although this cannot be considered a purely clear sky.

[Figure]

Figure 1. Examples of daily mean TCWV over tropical land obtained from CanESM5 with different cloud mask (pixels with cloud fraction larger than 75%, 50%, 25%, and 5% at all pressure levels are considered as clouded) along with data from TCWV-COMBI and ERA5, July 1st 2003.

Specific comments:

3. L114: How sensitive are the results with respect to these screening thresholds?

Reply: We have used the CanESM5 model as an example. The two figures below show the normalized PDFs of TCWV (left) and ω500 (right) for different cloud threshold tests: for these PDFs, the data with cf > 75%, > 50%, > 25% and > 5% are considered as cloudy pixels and are removed. The figures show that on the one hand, although most of the TCWV reside in the range 5 ~ 15 kg/m$^2$ whatever the threshold was set, the moist tail decays strongly with stricter cloud masks. On the other hand, the PDFs of corresponding ω500 don't change as much as the TCWV, with a slight shift of the sampled atmospheric situations towards more subsiding cases.

[Figure]

Figure. Normalized PDFs of TCWV and ω500 from CanESM5 data over tropical land area with different cloud masks.

To further discuss the sensitivity of the cloud mask, the TCWV obtained from CanESM5 with different cloud masks were then sorted according to the corresponding ω500 bins (10 hPa/day). The TCWV-COMBI and ERA5-TCWV are also included for a reference. As shown in the figure below, the TCWV at each ω500 bins are lower when the cloud mask is stricter, as you underlined, the results are dependent on the cloud-screening process. Therefore, normalization process is employed to analyse the time evolution with respect to the mean state. Nevertheless, the result from CanESM5 data with a threshold on cf < 50% is comparable to the ERA5 TCWV and presents a dry bias comparing to TCWV-COMBI.

[Figure]

Figure. The mean TCWV from CanESM5 (with different cloud masks), TCWV-COMBI (red line), and ERA5 (green line) in different circulation regimes of ω500 over land.

The following comments has been added in the manuscript:

L12: Since the results are sensitive to the scene selection applied in the data process, discrepancies are observed among the datasets. Therefore, normalization process is employed to analyse the time evolution with respect to the mean state.

L111: While the results are dependent to the cloud masks, normalization process is employed to better analyse the time evolution with respect to the mean state instead of strict biases.

4. Table 2: Why Eyring et al. (2016) as reference for ERA5? Wouldn't Hersbach et al. (2020) be more appropriate?

Reply: Thanks for the observation. We agree that the Hersbach et al. (2020) are more appropriate, we have revised the manuscript and fixed the problem.

5. Figure 2: Why is there this change in variability in TCWV-COMB less variance in the later period.

Reply: The change of variability is due to the change of instrument over the period. As discussed in section 2.1 in the manuscript, the TCWV-COMBI dataset merged satellite-based data records of MERIS (from 2002 to 2012), MODIS (from 2011 to 2017) and then OLCI (from 2016 to 2017, which is out of the scope in this research) over land. The stability of the CCI dataset has been thoroughly evaluated in the "Product Validation and Intercomparison Report", which should be available soon on the ESA CCI Water Vapour web page (https://climate.esa.int/en/projects/water-vapour/key-documents/). However, because the report is not yet accessible, we have gathered below some results that are relevant here:
1) by looking at the missing data in MERIS, MODIS, and OLCI periods, it's clear that the MODIS cloud mask is less strict than the cloud masks for MERIS and OLCI. The increased sample size after 2011 will lead to less variance of the daily mean value during this time.
2) validation of the dataset against GRUAN and SuomiNet show that the bias and corrected RMSD (cRMSD) are generally small and within ±1.5 kg/m$^2$ and 2.5 kg/m$^2$, respectively. Indicating that the dataset is accurate and stable for the whole observation period.

[Figure]

Figure. Time series of bias and corrected RMSD (cRMSD) between TCWV-COMBI dataset and the intercomparison and reference dataset for global land surfaces. The dashed vertical lines mark changes in the NIR observing system. The sample size contains the number of valid daily observations in a month (middle panel). Figures taken from the "Product Validation and Intercomparison Report" (Falk et al, 2022, ESA Water Vapour Climate Change Initiative)

We have revised Figure 2 to highlight the changes in the instrument and the following sentences have been added in the manuscript to reflect this point:

[Figure]

**Figure 2.** Time series of daily mean TCWV in the tropics (30∘S - 30∘N) over (a) land areas under clear-sky condition and (b) ocean areas except for heavy precipitation (see details in Section 2). The time series covers the period 07/2003-12/2014. The gray lines denote the individual CMIP6 models while the black line represents their ensemble mean. The green line represents ERA5 and the red line is the ESA CCI_WV TCWV-COMBI.

L180: Although there are changes in variability in TCWV-COMBI data over tropical land with the inclusion of MODIS since 2011, validation of the dataset against GRUAN and SuomiNet show that the dataset is stable and accurate during the whole observation period.

6. Figure 2: Are the outliers in Tropical-land TCWV-COMBI realistic (especially in 2004)? I doubt that TCWV can almost double from one time to the other. I recommend to include a cautionary note.

Reply: Thanks for this comment. The daily mean TCWV are the mean of the available clear-sky sample observed during that day, therefore, the value is dependent on the cloud mask. Here we take a closer look at the highest daily mean TCWV from the TCWV-COMBI data record (43.27 kg/m$^2$, observed on 9 June 2004). As shown in the figure below, the data are distributed over Indian, and most of the valid data reside between 45 ~ 55 kg/m$^2$. The day before (8 June 2004), on the other hand, has observations all over the tropics and most of the valid data are located around 10 ~ 20 kg/m$^2$. The differences in available sample size under the clear-sky condition each day will lead to a considerable variation in the daily mean value on two consecutive days.

[Figure]

**Figure**. Distribution map of TCWV (first row) obtained from TCWV-COMBI dataset on 8 June 2004 and 9 June 2004 and the normalized PDF of the TCWV.

The following sentence has been added in the manuscript to address this point:

L164: Overall, the different data records agree well with each other despite some 'outliers' observed over tropical-land in TCWV-COMBI data that resulted from the differences in available sample size under the clear-sky condition each day.

8. L186: I'd recommend to present the final sentence of this subsection ("The bimodal distributions ..."), explaining the cause of the existence of bimodality, earlier in this section (e.g., in L176).

Reply: We thank the reviewer for the comment. We have revised our paper as suggested.

9. Figure 3: It could be mentioned already regarding Fig. 3 that the observations show higher frequency of moist extremes.

Reply: We thank the reviewer for the comment which is now added into the description of Figure 3.

10. Figure 4: I'd prefer some color scheme centered around 1 such that 1 is represented by white, similar to the one used in Fig. 7. This would make the anomalies clearer. (Similar for Figs. 5/8/10).

Reply: Thanks for the comments. We have revised the figures in question as suggested.

[Figure]

**Figure 4.** Normalized percentiles of the TCWV over land areas for every data record. The percentiles are grouped into bins of 10% intervals. The x-axis represents the percentiles intervals, and the y-axis represents the year. Note that the period starts in 2004 instead of 2003 to focus on full years.

[Figure]

**Figure 5.** Same as Figure 4 but for oceans.

[Figure]

**Figure 8.** Normalized TCWV with respect to the 2004 – 2014 mean over tropical land areas at each dynamical intervals (ω500) in 10 hPa/day computed from each data record.

[Figure]

**Figure 10.** Normalized TCWV with respect to the 2004-2014 mean over tropical ocean areas at each dynamical intervals (ω500) in 10 hPa/day computed from each data record.

11. L222 and Figure 6: I found the description of Fig. 6 here somewhat oversimplified. E.g. over trop. ocean the TCWV-COMBI data set shows moistest regimes not for upward motion, but for weak subsidence. I was wondering about the reasons, and also think this should at least be mentioned and discussed in the text. Also, CMIP6 and ERA5 seem to agree somehow better in terms of TCWV(w500), than they compare to observations - any ideas for possible reasons?

Reply: Thanks for the comment. The evaporation from the oceans is the primary source of water vapour in the atmosphere, the oceanic boundary layer is humid even in weak subsidence regime.
For the second comment, concerning the w500, the agreement is better because for CMIP6 and ERA5, the TCWV were sorted by the corresponding w500 of the data record. As for TCWV-COMBI data, since there is no observation on the w500, the w500 from ERA5 is employed as the reference.

L225: The evaporation from the oceans is the primary source of water vapour in the atmosphere, the oceanic boundary layer is humid even in weak subsidence regime. Besides, as the reference $\omega 500$ for TCWV-COMBI are from ERA5, it is sensible that there are differences observed in the TCWV-COMBI data as it stays humid in both upward and weak subsidence regions.

12. L254 and Figure 8: Interestingly, all data sets except ERA5 show a dry anomaly around 2008. Any idea why ERA5 disagrees in that respect? Perhaps the formulation in L254 should be more careful, saying that the 2008 dry anomalies for ERA5 are only observed for subsidence and weak upward motion regimes.

Reply: We agree that the dry anomalies for ERA5 only observed for subsidence and weak upward motion regimes in 2008. As shown in Figure 2,3, and 7, the ERA5 data have a dry bias comparing to the other data records. The detailed comparison of water vapour in the reanalysis, radiosonde and satellite data further indicate that the reanalysis data slightly underestimated the water vapour in 2009/10 winter (Du, Minkang, et al. "Water vapour anomaly over the tropical western Pacific in El Niño winters from radiosonde and satellite observations and ERA5 reanalysis data." Atmospheric Chemistry and Physics 21.17 (2021): 13553-13569.). The underestimation can lead to the dry bias of climatology used to calculate the anomalies, which would, in turn, lead to the discrepancy observed in ERA5.

The following sentence have been added in the manuscript to clarify this point:

L250: Although different patterns are observed, dry anomalies occurred in 2008 in all the data records (except for ERA5 (Du et al., 2021), where dry anomalies were observed for subsidence and weak upward motion regimes), and wet anomalies occurred in 2010.

L292: I think this should read "moistening trend" here, if it refers e.g. to Fig. 7. L312: I'd add: ... and the GCMs over oceans.

Reply: Thanks for the comment. The manuscript has been revised as suggested.

L317: Maybe better: "This clearly shows the necessity/importance of having access to the simulated water vapour ..."

Reply: We appreciate the comment. The manuscript has been revised as suggested.

Technical corrections:
L11: absent in?
L16: a critical role
L64: investigateS
L64: One "the" too much.
L65: participated in?
L67: Blank after "datasets".
L82: is discussed
L95: participating in?
L127: considerable --> considerably
L129: ERA5
L130: Comma after "Hence".

L136: approches --> approaches
L137: evaluateS
L161: Blank after "TCWV".
L163: Better "time series" instead of "climatology"?
L163: Fig2a --> Fig. 2a (Please check throughout the paper that references to figures are according to ACP style.
Fig. 2, caption: cover
L193: reddish
L233: Blank after "atmosphere".
L238: Blank after "different".
L240: Blank after "models".
L247: regime
L249: interval
L259: Blank after "variabilities".
L291: tipping
L291: associated with
L296: associated with
L298: ...ESA TCWV-COMBI data and ERA5... L303: Blank after "factors".
L307: differences
L314: I think a parameter can not be "easier".

Reply: Thank you very much for the above observations. The manuscript has been revised as suggested.

**Anonymous Referee #3**

This study evaluates the fidelity of model-simulated and reanalysis-produced total column water vapor over the tropics using the ESA TCWV-COMBI data as a benchmark. Given the role of the water vapor feedback in amplifying greenhouse gas-induced warming, it is important to monitor the variability and change of water vapor on multiple time scales and to examine whether the physical processes governing the variability and change are accurately depicted in global climate models. However, there are some issues in the manuscript which the author need to clarify.

The main objective of this study is to evaluate tropical water vapor simulated from CMIP6 models under the AMIP configuration using the ESA TCWV-COMBI data as a benchmark. While this implies that the ESA TCWV-COMBI data are accurate and reliable, the main text seems to indicate the presence of temporal discontinuity (and also spurious trends) in the data set, especially, over the land regions. So I think the author first need to demonstrate the accuracy and stability of the dataset by comparing it with other independent data sets such as intercalibrated UTH datasets.

Reply: We thank the reviewer for the comment. The stability of the TCWV-COMBI dataset has been evaluated in the "Product Validation and Intercomparison Report" (will be available soon on https://climate.esa.int/en/projects/water-vapour/key-documents/). Validation of the dataset against GRUAN and SuomiNet show that the bias and corrected RMSD (cRMSD) are generally small and within $\pm1.5$ kg/m$^2$ and 2.5 kg/m$^2$, respectively. It is reasonable to conclude that the TCWV-COMBI is stable and accurate.

[Figure]

Figure. TCWV-COMBI data and ERA5 over global land surfaces (top panel), between TCWV-COMBI and merged microwave over global ice free ocean surfaces (middle panel) and between TCWV-COMBI and ERA5 over global surfaces (bottom panel)

We would like to point out that UTH focuses of high altitude while the TCWV is influenced by near surface layers. Detailed comparison of UTH datasets is beyond the scope of this research. Therefore, we didn't conduct comparison on the UTH. However, we agree that studies on independent datasets are valuable, we will study them in the future work.

The following sentence has been added in the manuscript to address this point:

L85: Validation of the dataset against GRUAN and SuomiNet show that the bias and corrected RMSD (cRMSD) are generally small and within ±1.5 kg/m2 and 2.5 kg/m2, respectively.

The discrepancies in TCWV between the ESA data and GCM/ERA5 are attributed in part to clear-sky sampling issue. I am wondering if the cloud-screened modeled and ERA5-produced TCWV is substantially different from the corresponding raw output (i.e., the scene selection is not applied). Please present some plots showing the difference. In addition, given the large difference in the horizontal resolution between GCMs ( > ~1 deg) and the ESA dataset (0.05 deg), I am not sure whether the scene selection method described in the text is a suitable way.

Reply: We appreciate this comment. As discussed in our reply of R1C2 and R1C3, the results are dependent on the cloud screening process. By adapting the scene selection criteria of the Sohn and Bennartz (2008) study, ERA5 data with tcc < 95% and tclw < 0.005 kg/m$^2$ over land are retained for the following evaluation analysis. Besides, a series of threshold tests were conducted for CMIP6 models to get comparable clear-sky data with ERA5 and TCWV-COMBI data.

Figure 2 indicates a much larger discrepancy between the data sets over the land than over the ocean. This might be caused by potential errors of IR-based TCWV. So please show that the IR-based TCWV is in good agreement with MW-based TCWV over the tropical ocean.

Reply: Thanks for the comment. Comparison between TCWV-COMBI and merged microwave over ocean and between TCWV-COMBI and AIRS over global surfaces are discussed in the "Product Validation and Intercomparison Report" (will be available soon on https://climate.esa.int/en/projects/water-vapour/key-documents/). As shown in the figure captured from the report, the data exhibit a high level of correlation ($R^2 \geq 0.99$) with the bulk of the data pairs being close to the one-to-one line.

[Figure]

Figure. Scatter plots of daily TCWV-COMBI against TCWV of merged microwave (top) and AIRS (bottom) for ice-free ocean and global surfaces, respectively.

Although I agree with the authors that TCWV is influenced by large-scale atmospheric circulation over the tropics, the underlying surface temperature is also a major factor determining the magnitude of TCWV. However, this aspect is not accounted for in the analysis from Fig. 6 to Fig. 10 in which some panels exhibit physically unreasonable features. So I think the author first remove the thermodynamic component before analyzing the relationship between TCWV and large-scale atmospheric circulation.

Reply: We thank the reviewer for pointing out this interesting aspect. We agree that the TCWV is influenced by the SST. The CMIP6 models that are evaluated under the AMIP scenario with the same prescribed SST. However, the goal of this paper is to evaluate the TCWV data records from the aspect of large-scale circulation. While we agree that there are discrepancies observed among the data records, especially over land area, we would like to point out that the TCWV-COMBI data and ERA5 data vary within the ensemble spread of CMIP6 models, indicating that the mean models could correctly represent the evolution of water vapour with respect to large-scale circulation. Nevertheless, we believe that analysis with consideration of the influence from SST is important and we will study it in a more detailed manner in the future.

L5: I am not sure whether the authors examined the "evolution" of the large-scale circulation in the manuscript. I think the authors just analyzed the interannual variability of TCWV as a function of 500-hPa vertical velocity.

Reply: Thanks for the comment. The intercomparison is conducted by decomposing the TCWV into dynamical regimes defined from the vertical velocity for each year. The temporal evolution of the TCWV and ω500 are revealed in the revised figure 7 to figure 11. We have revised the sentences in question for clarification purpose:

L5: The intercomparison is performed according to the probability density function (PDF) of the total column water vapour (TCWV), as well as its evolution with respect to large-scale overturning circulation.

L20: surface temperature "from" robust thermodynamical constraints – "via", "through" or other word instead of "from"?

Reply: Thanks for the observation. We have revised the sentence in question as follows:

L20: The precipitable water, mainly concentrated in the atmospheric boundary layer, is directly influenced by the surface temperature through robust thermodynamical constraints. The concentration of boundary layer water vapour will increase up to 7% /∘C globally, confirmed by simulations and observations (Allan et al., 2014).

L56: "both the complementarity between the sensors" and what?

Reply: Thanks for the observation. We have revised the sentence in question as follows:

L56: This intercomparison not only highlighted the complementarity among the sensors, but also underlined the caveats in the studies of trends and variabilities induced by artificial break points contained in the CDRs, such as calibration changes, retrieval algorithms, resolution changes that impact the sampling, etc..

L95: A subset of "seven" GCMs – I think the seven GCMs analyzed in this study are a subset of CMIP6 models.

Reply: Thanks for this remark. The text has been corrected as follows:

L95: Seven GCMs participating to CMIP6 are evaluated…

L97: ESGF instead of ESFG?

Reply: The manuscript has been revised as suggested.

L99: at the model vertical resolution – Do the ESGF websites provide model output at the model vertical resolution?

Reply: The data of CMIP6 models are obtained from the node of IPSL. The output of RH is at the model vertical resolution.

L151-152: "The TCWV data are sorted upon 10 hPa/day-bins of monthly values of w500" – The occurrence frequency could be substantially different across the bins, as shown in Fig. 6.

Reply: The PDFs of ω500 show a strong maximum for ω500 around 20 ~ 30 hPa/day, indicating that the rate of subsidence of the free troposphere air is primarily constrained by the clear-sky radiative cooling rate of the atmosphere (Bony et al., 2004). By decomposing the TCWV by the vertical motion of atmosphere, we could focus on the evolution of a particular regime. The biases in the simulation of the dynamical patterns will not affect the model-data comparison (Brogniez and Pierrehumbert, 2007). Besides, we have revised the units in Fig. 6.

L163: the climatology – I don't think Figure 2 shows the climatology.

Reply: We understand the remark and we have corrected the manuscript as follows:

L163: Figure 2 shows the time series of the TCWV of the different datasets over land…

L164: Please specify in which aspects the datasets agree with each other, as there are distinct mean biases between the data sets.

Reply: We have specified as follows:

L164: Range of daily mean TCWV exists some differences, but all datasets have the same changing tendency.

L166: a very weak interannual variability – Does this mean that the ENSO impact is small?

Reply: Figure 2 shows the time series of daily mean TCWV of in the tropics over lands and oceans. As discussed in R1C2 and R1C6, the value is affected by the cloud-screen process. Therefore, the impact of ENSO is not apparent by reviewing the time series. The following comments has been added in the manuscript:

L166: Since the results are dependent on the cloud-screen process, the detailed discussions on the impact of ENSO events are discussed in following section 4.2.

L168-173: I failed to understand these sentences. Given that the ESA dataset was constructed by excluding cloud-contaminated observations, the ESA TCWV is likely to be drier than both model simulations and the ERA5 data. Please clarify this inconsistency.

Reply: The data over ocean areas measures TCWV under all-weather conditions except for heavy precipitation, therefore the results over ocean areas appears to be moister comparing the results over land areas. Besides, the TCWV-COMBI data is moister than both model simulations and the ERA5 data over both the land and ocean areas. Nevertheless, as discussed in R1C2 and R1C6, the results are dependent with the cloud-screening process. Therefore, the TCWV data are decomposed into dynamical regimes to analyse the evolution of a particular regime. We have revised the manuscript to clarify this problem:

L168 - 173: More specifically, the ESA TCWV-COMBI data is moister than ERA5 and most CMIP6 models over both the land and the ocean areas, and this moist bias is even more pronounced over tropical land (Fig 2a, $\sim$ 10kg/m$^2$ over land vs $\sim$ 2kg/m$^2$ over ocean). On the other hand, the daily mean values of water vapour concentration over ocean areas are higher than the values over land areas. This difference can be explained because the TCWV datasets over land areas are composed of clear-sky-only data, which are likely drier than the nearby cloud area for a given location and thus translates into a dry bias associated to moistening processes by convective clouds (Sohn et al., 2006). Hence, the cloud screening over land makes it difficult to compare the datasets directly.

L173-174: Please clarify this sentence.

Reply: We have clarified as follows:

L173: In addition, since the boundary layer is drier in the continental subtropics and the maritime stratocumulus zones are wetter at low levels, the ocean areas should appear moister than the land areas.

L180-182: Please clarify the inconsistency.

Reply: The cloud screening process applied for CMIP6 models and ERA5 aims to remove cloudy pixels, with due consideration to maintaining enough data for further evaluation. As discussed in R1C3, the moist tail of the simulation model decays strongly with stricter cloud masks. Therefore, the results are dependent with the cloud-screening process. We have revised the manuscript to clarify this problem:

L180: Indeed, the results of simulation models and ERA5 are dependent on cloud conditions, consequently will lead to differences in the comparison.

L184: situations – I am not sure what "situations" mean here.

Reply: We have clarified as follows:

L184: While the main peak of PDF is nearly identical for TCWV-COMBI, ERA5 and CMIP6, there is a divergence for the secondary peak.

L199: It doesn't seem that in ERA5, the year 2008 exhibits a distinct dry anomaly over all the TCWV range.

Reply: We have revised the sentence in question for clarification purposes:

L199: Overall, anomalies are observed in the time period for all data records: 2008 appears as a dry year while 2010 reveals a clear signal of humidification, especially over the high parts of the distributions of TCWV (percentiles > 60%) of TCWV.

L207: the CMIP6 models "that" are evaluated under the AMIP scenario – remove "that"?

Reply: The manuscript has been revised as suggested.

L209: concerned – what does "concerned" mean here?

Reply: The sentence has been revised as follows:

L209: Hence this explains that anomalous years are the results of El Nino Southern Oscillation…

L211: Please clarify this sentence.

Reply: The sentence has been revised as follows for clarification purposes:

L211: 2008 and 2011 are characterized by a very negative ENSO index, while 2010 is an intermediate year, which starts with a positive ENSO cycle and is followed by a negative one.

L224: large-scale ascent is globally associated with a moister troposphere – The largest TCWV values are found over a weak subsidence regime for ERA5.

Reply: The sentence has been revised as follows for clarification:

L224: large-scale downward motion is associated with a dry troposphere, while large-scale ascent is associated with a moister troposphere (except for ERA5 over land, where the most humid regimes are over weak subsidence).

L228: difficulties – what does "difficulties" mean here?

Reply: We have revised the sentence for clarification purposes as follows:

L228: the discrepancies in the TCWV reveal difficulties in representing the moistening processes of the tropical atmosphere

L230-231: I don't think the authors demonstrate this point in the manuscript.

Reply: The large-scale downward motion is associated with a dry troposphere, and large-scale ascending motion is associated with humid atmosphere (Bony et. al., 2004; Brogniez and Pierrehumbert, 2007). However, as the results in Figure 6 (c) show, the moistest regime of TCWV from ERA5 occurred over a weak subsidence regime instead of the strong ascending regime. This is partly because the results are dependent on cloud masks. Therefore, it's judicious to conclude that the large-scale advection humidification/drying processes are not accurately represented. We have revised the following sentences for clarification purposes:

L230-231: It is worth mentioning that the moistest regime of TCWV from ERA5 over land areas occurred in a weak subsidence regime instead of the strong ascending region. This difference is partly because of the cloud screening processes. Therefore, the results could not accurately represent the large-scale advection humidification/drying processes.

L236-237: while the most ... with the moister troposphere (blue) – I don't agree with the authors. Please check ERA5, CanES5, CESM2, and CESM2-WACCM.

L235-238: As shown in Figure 7, all the data records (except for ERA5) agree that driest troposphere (red) are associated with the most positive ω500 bins (meaning the areas of highest downward motion). The moistest troposphere, however, are not always located in the most negative ω500 bins (the highest upward motion). ERA5 and the CMIP6 ensemble mean …

L243-244: Please provide more detailed information on "strong effective climate efficiency".

Reply: The CanESM5 has an increased effective climate sensitivity comparing to all current CMIP6 models (Virgin et.al., 2021). The positive low and non-low shortwave cloud feedbacks – particularly with regards to low clouds across the equatorial Pacific, as well as subtropical and extratropical free troposphere cloud optical depth – are the dominant contributors to CanESM5's increased climate sensitivity. The following comment has been added for clarification purposes:

L243-244: The discrepancy observed from CanESM5 is partly because of its strong effective climate efficiency compared to all current CMIP6 models, as the positive low and non-low shortwave cloud feedbacks – particularly with regards to low clouds across the equatorial Pacific, as well as subtropical and extratropical free troposphere cloud optical depth – are the dominant contributors to CanESM5's increased climate sensitivity (Virgin et al., 2021).

L245: the transition dry/moist occurs – the transition of the dry/moist regime occurs?

Reply: We have revised the manuscript and fixed the problem.

L246: occurs around 60 hPa/day – I am not sure whether this statement is consistent with other figures in the manuscript.

Reply: We have removed this statement.

L250: wet anomalies occur in 2010 for all of the data records – Moist anomalies are not distinct for the ESA data set, ERA5, and IPSL-CM6A-LR.

Reply: We have revised the manuscript as follows:

L250: Although different patterns are observed, dry anomalies occurred in 2008 in all the data records (except for ERA5 (Du et al., 2021), where dry anomalies were observed for subsidence and weak upward motion regimes), and wet anomalies occurred in 2010.

L252-255: Please clarify the description.

Reply: We have revised the sentence for clarification purposes as follows:

L252: The ESA TCWV-COMBI record reveals a clear moistening tendency, especially over the subsiding branch of the atmospheric circulation after 2011. One of the major causes of the turning point is the inclusion of MODIS data since 2011 which would increase the sampling size of the data and in turn affect the tendency.

L259-261: Please reword the sentence.

Reply: We have revised the sentence for clarification purposes as follows:

L259: Finally, the CMIP6 models and their ensemble mean show consistent interannual variabilities: dry anomalies occurred in 2008 in all the data records (except for ERA5 (Du et al., 2021), where dry anomalies were observed for subsidence and weak upward motion regimes), and wet anomalies occurred in 2010.

L263: The verb is missing?

Reply: The sentence in question has been revised as follows:

L263: Oceanic situations were also discussed with respect to the large-scale circulation.

L265-266: Please clarify the sentence.

Reply: We have revised the sentence for clarification purposes as follows:

L265-266: All data records, except for TCWV-COMBI, CanESM5 and IPSL-CM6A-LR, show that the strongest ascending zones are corresponding to the very humid regions.

L268-271: Please clarify these sentences.

Reply: We have revised the sentence for clarification purposes as follows:

L268-271: Since the transport model in the boundary layer of IPSL-CM6A-LR affects both shallow and deep convection regimes, a compensational bias would be induced, thus a moist bias will be present in the weak upward motion regimes (~ -30 hPa/day).

L272: I don't think Figure 10 shows temporal tendencies.

Reply: We have revised the following sentence for clarification purposes:

L272: The normalized TCWV with respect to dynamical intervals over ocean areas are shown in figure 10.

L292: the drying trend being present for all regimes of w500 – inconsistent with L290.

Reply: The following sentence has been revised for clarification purposes:

L292: dry anomalies occurred in 2008 in all the data records (except for ERA5 (Du et al., 2021), where dry anomalies were observed for subsidence and weak upward motion regimes), and wet anomalies occurred in 2010.

L301-302: I don't think the authors demonstrate this aspect in the manuscript.

Reply: The following sentence has been revised for clarification purposes:

L301-302: There are discrepancies observed among the data records, probably caused by the lateral mixing, outflows from clouds, and the precipitation efficiencies of the convective schemes.

L310: the use of AMIP scenario, defined from prescribed sea surface temperatures – please clarify the sentence.

Reply: The sentence in question has been revised for clarification purposes:

L310: the CMIP6 models under AMIP scenario (with prescribed sea surface temperature), and ..

L314: water vapour seems to be an easier parameter than cloud – please reword the sentence.

Reply: We have revised the sentence as follows:

L314: even though it's easier to compare water vapour than clouds.

L315-317: Please reword the sentences.

Reply: The sentences in question have been revised as follows:

L315-317: The water vapour profiles (and sometimes the integrated values) from climate models usually have coarser spatial resolution than satellite observations. The satellite measurements, on the other hand, are often strictly restrained by cloud contamination.

Table 2: It is unlikely that Eyring et al. (2015) is an adequate reference for the ERA5 dataset.

Reply: We have revised the manuscript and fixed the problem.

Figure 1: The TCWV maps have some land regions where the data are missing. Why is that?

Reply: The missing data are land areas with cloud and ocean areas with heavy precipitation.

Figure 6: The units are incorrect in the figure.

Reply: We have revised the figure to fix this problem:

[Figure]

Figure 7: The distributions for the ESA and ERA5 datasets are inconsistent with Fig. 6c.

Reply: We would appreciate if the reviewer could provide more precise examples of the inconsistency between figure 6c and figure 7. We would like to point out that both figures indicate that the ESA data have a wet bias and ERA5 data have a dry bias comparing to the simulation models.

There are typos and grammatical errors in the manuscript. Please correct them.

Reply: We have checked the manuscript. We hope that this is now improved.

---

## Author Response (AR2)

**Author's Response**

Paper Number: ACP-2021-976
Paper Title: Evaluation of tropical water vapour from CMIP6 GCMs using the ESA CCI "Water Vapour" climate data records

We thank the editor and the anonymous referee for the comments on our manuscript that help to improve the quality of this manuscript. We have carefully addressed all the points raised by the reviewer. Please find below our response and corresponding changes made in the revised paper.

Sincerely,
On behalf of all authors,
Jia He

**Anonymous Referee #3**

Focusing on the tropical belt, this study compares interannual variability of total column water vapour of global climate models and ERA5 with that inferred from the ESA TCWV-COMBI data set. Large discrepancies between ESA TCWV-COMBI and ERA5/GCMs are found over the land, and the authors attribute the discrepancies over the land to a clear-sky scene selection issue as well as deficiencies in model parameterizations. This is an interesting study, but I think there are some aspects that need to be clarified.

Data sampling issue: I agree with the authors that discrepancies over the tropical land are mainly related to clear-sky scene selection. Figure 1 implies, however, that the discrepancies may also result from incomplete spatial/temporal coverage of satellite observations. For instance, in the bottom left panel of Fig. 1, TCWV values are missing over some land regions (e.g., 20E-30E, diagonal direction from southwest to northeast) because those regions are not covered by satellite observations, rather than due to cloud contamination. In addition, Table 1 indicates a discrepancy in temporal sampling between ESA TCWV-COMBI (day-time only) and ERA5/GCMs. Given this sampling issue, I think monthly-mean data (or weekly-mean) are more appropriate for this intercomparison study than daily-mean data.

Reply: We appreciate this comment. The reason that we conduct this evaluation analysis using daily data instead of monthly mean for several reasons. Firstly, our intention is to evaluate the datasets with the highest temporal resolution, as the temporal averaging will mask out the extremes and the PDFs would have been smoothened. Secondly, since the cloud condition varies significantly over short time scales, quantification at high temporal resolution is required. Last but not least, the spatial resolutions for GCMs are much coarser than the observation data of TCWV-COMBI. To maintain certain amount of data for further evaluation after cloud screening process, it is judicious to use the daily data in our analysis. Moreover, the datasets are decomposed into dynamical regimes to evaluate the evolution of a particular regime, the discrepancy result from the swaths gaps of satellite observations will not affect the model-observation comparison.

Potential mismatch in large-scale atmosphere circulation between daily and monthly time scales: In this study, the mean climatology and interannual variability of daily-mean TCWV are examined as a function of monthly-mean 500-hPa vertical velocity. However, due to large day-to-day variability of vertical motion, descending motion can occur for some days of a given month while large ascending motion is prevalent over that month, and vice versa. This mismatch in vertical velocity between daily and monthly time scales can result in potential errors and inconsistencies. In other words, low values of daily-mean TCWV accompanied by daily-mean descending motion could be assigned into ascending motion bins at monthly time scale. In fact, the distribution of ESA TCWV as a function of monthly-mean 500-hPa vertical velocity shown in Fig. 7d appears to be inconsistent with Fig. 2: while the largest values of ESA TCWV are found over weak subsidence regime in Fig. 7d, Figure 2 suggests a clear

linkage of humid regimes with large ascending motion. So I think that the analysis associated with Figs. 7-11 should be conducted at the same time scale using either daily or monthly-mean data.

Reply: Thanks for the comment. The previous study suggests that the ω500 is sensitive to local dynamics and subject to significant biases at the instantaneous scale (Trenberth, K.E., Stepaniak, D.P. and Caron, J.M., 2000. The global monsoon as seen through the divergent atmospheric circulation. Journal of Climate, 13(22), pp.3969-3993). The monthly vertical motion can represent a mixture of ascending and descending atmospheric conditions. By adopting the monthly mean of ω500 in our evaluation, the fluctuations of shorter time scales, where small-scale convection probably dominates, are ignored. However, research show that the ω500 with shorter time scales are unreliable (Höjgård-Olsen, E., Brogniez, H. and Chepfer, H., 2020. Observed evolution of the tropical atmospheric water cycle with sea surface temperature. Journal of Climate, 33(9), pp.3449-3470). Therefore, the daily water vapour data are decomposed by the monthly mean ω500 in our analysis.

Figure 2 shows that the large-scale downward motion is associated with a dry troposphere, and large-scale ascending motion is associated with humid atmosphere, while the in Figure 7d, the results show that the atmosphere remains humid even in the weak subsidence regime, this was because that the evaporation from the ocean is the primary source of water vapour in the atmosphere, the oceanic boundary layer is humid even in weak subsidence regime.

L7: "especially" – Please consider deleting it as the second part of the sentence describes the ocean case.

Reply: We have revised the manuscript as suggested.

L9: El Niño/La Niña

Reply: We have revised the manuscript as suggested.

L35: total water vapour content (TCWV) – total column water vapour (TCWV) according to L65

Reply: We have revised the manuscript as suggested.

L125: It is unclear whether the same method is applied to the ocean case as in the land case.

Reply: The data over land areas are under clear-sky condition, while the data over ocean areas are under all-weather condition except for heavy precipitation. The following sentence has been revised to clarify this problem:

"Over tropical oceans, the percentage of data that remained after removing the pixels under heavy precipitation conditions range from 99.79% to 99.98."

Figure 1: Some values are plotted over some regions in the southeast Pacific (~20S, 100W) and south Atlantic (~20S, 5W). Are these regions big islands?

Reply: The land-sea mask applied for the sampled model in Figure 1 (CanESM5) was defined by the land area fraction product of the model. Here we defined the area as land where the percentage of the grid cell occupied by land larger than 50%. The areas in question are archipelago, for instance, the Isla de Pascua (~30S, 110W) and the Saint Helena (~20S, 5W).

L118: "The land area fraction product is adopted as the land-sea mask for the CanESM5 model. Here we defined the area as land where the percentage of the grid cell occupied by land is larger than 50%."

L171-172: There are differences in the range of daily mean TCWV exists some differences – Please clarify.

Reply: We have revised the manuscript as follows:

"There are differences in the range of daily mean TCWV."

L172: all datasets have the same changing tendency – I failed to understand the meaning of "changing tendency".

Reply: We have revised the manuscript as follows:

"all datasets varied with the seasons"

L177: softer – weaker?

Reply: We have revised the manuscript as suggested.

L178: interannual signal – interannual variability?

Reply: We have revised the manuscript as suggested.

L180: Given that both in situ and satellite observations are used for data assimilation processes in ERA5, it is very surprising to see a large discrepancy (~10 kg/m2) in TCWV between ESA TCWV-COMBI and ERA5. Do the raw ERA5 data without the cloud screening show a similar bias?

Reply: Thanks for the comment. The daily mean data of TCWV over the tropical land areas from raw ERA5 data under all-weather conditions (black), ERA5 under clear-sky conditions (green), and TCWV-COMBI (red) were displayed in the figure below. As the satellite observations are assimilated within the ERA5, the raw ERA5 data are indeed closer to the observations. For the clear-sky ERA5 results, we believe that the large dry bias is due to the fact that the cloudy area, which has been removed as contaminated data, is likely wetter than the nearby clear-sky area for a given location.

[Figure]

L195-196: This sentence is inconsistent with Fig. 4.

Reply: Thanks for the comment. We have removed this sentence to clarify this problem.

L197-198: Figure 3a shows distinct differences in the long-term mean and associated standard deviation of ESA TCWV between 2003-2011 and 2011-2015, which doesn't appear to support the argument that the ESA TCWV-COMBI dataset is stable and accurate during the whole observations period.

Reply: Thanks for the comment. The ESA TCWV-COMBI dataset is stable and accurate during the evaluation period in this research. The stability of the CCI dataset has been thoroughly evaluated in the "Product Validation and Intercomparison Report", which should be available soon on the ESA CCI Water Vapour web page (https://climate.esa.int/en/projects/water-vapour/key-documents/). As shown

in the following figure generated in the report, the ESA TCWV-COMBI data is stable during the observation period in this research (2003 ~ 2014).

[Figure]

Figures 4a/5: Do the resutls based on raw ERA5 and GCMs data (without cloud screening) show a similar distribution?

Reply: We have recalculated the PDFs of ERA5 and GCMs over tropical land under all-weather conditions and the results are displayed in the figure below. This figure has to be compared to Figure 4a. As shown in the figure, the raw datasets are more humid than the data under clear-sky only conditions, with a second peak near 50 kg/m$^3$ that reaches the about same amplitude than the peak at 10-15 kg/m$^3$. The cloud screening has removed some of the humid pixels and smoothened the second peak of the raw data.

[Figure]

Figures 5/6: It is difficult to understand why the land case shows large discrepancies between ESA TCWV-COMBI and ERA5 while the distribution is generally similar to each other for the ocean case.

Reply: Thanks for the comment. The TCWV-COMBI dataset merged the NIR water vapour products over land under clear-sky conditions and microwave measurements over the ocean under all-weather conditions. As there is no cloud mask for the models, several parameters on the cloud are adopted in the screening process. Although a series of threshold tests were conducted, there are still differences between the criteria applied in the cloud screening process and the cloud-mask of TCWV-COMBI, therefore, uncertainties are expected for the comparison over land.

L224: El Nino -> El Niño

Reply: We have fixed the problem.

L240: Could you explain the reason why the most humid regimes are over weak subsidence for ERA5?

Reply: As shown in Figure 4a, the ERA5 data over land show a dry bias in the humid area comparing to other datasets, results in a drier ascending area comparing to other models.

L244-245: Please clarify the sentence.

Reply: We have revised the sentence for clarification purpose:

"Besides, as the reference $\omega 500$ for TCWV-COMBI are from ERA5, it is sensible that there are differences observed in the TCWV-COMBI data comparing to other datasets that are decomposed by the corresponding model products."

L261-265: Please clarify the sentence.

Reply: We have revised the sentence for clarification purpose:

"The discrepancy observed from CanESM5 is partly because of its strong effective climate efficiency compared to other CMIP6 models (Virgin et al., 2021). For the CanESM5, the positive low and non-low shortwave cloud feedbacks, as well as the subtropical and extratropical free troposphere cloud optical depth, particularly with regards to low clouds across the equatorial Pacific, are the dominant contributors to its increased climate sensitivity (Virgin et al., 2021)."

Figure 8: It is very surprising that convectively active regimes have very low TCWV values for ERA5 and CanESM5. In contrast, Figure 1 shows that large TCWV values are observed over those regions, although the analysis time period is different between the figures.

Reply: As shown in Figure 3a, the ERA5 has dry bias comparing to other models and TCWV-COMBI data under the clear-sky conditions over tropical land. This may result from the cloud screening process that has remove some of the humid pixels as contaminated by cloud. It is sensible that the overall value of TCWV at the circulation regimes are drier comparing to other models.

Figure 10: The distribution for CanESM5 appears to be inconsistent with the results shown in Fig. 7. In addition, the TCWV values appears to be too low compared to Fig. 3 where the lowest values of ocean-mean TCWV are much greater than 20-25 kg/m2.

Reply: Thanks for the observation. We have checked the results and revised the figure to fix this problem.

[Figure]

There are some grammatical errors in the manuscript. For instance, "The algorithm for NIR imagers [are] discussed (L82).

Reply: We have checked the manuscript. We hope that this is now improved.

---

## Author Response (AR3)

**Author's Response**

**Paper Number: ACP-2021-976**
**Paper Title: Evaluation of tropical water vapour from CMIP6 GCMs using the ESA CCI "Water Vapour" climate data records**

We thank the editor for the comments on our manuscript. Please find below our response and corresponding changes made in the revised paper. We hope that we satisfyingly addressed the comment, and that the manuscript will be now suited for publication.

Sincerely,
On behalf of all authors,
Jia He

**Editor**
Comments to the author:
Dear Jia He and co-authors,

your paper is accepted for publication in ACP with a minor point to be discussed.
I have a last question/recommendation: regarding the last comments of Ref. #3 on the data sampling issue and potential mismatch in large-scale atmospheric circulation between daily and monthly time scales, I see no changes in the manuscript - though these seem to be points of discussion - but also no argument why this was not done? You might provide an argument, or, what I would recommend, briefly address these potential errors in the manuscript.

Reply: Thank you very much for this comment. We have now added the following paragraph in the manuscript for clarification purpose:

L72: The daily water vapour data and monthly mean $\omega 500$ are adopted in our analysis for several reasons. Firstly, our intention is to evaluate the datasets with the highest temporal resolution, as the temporal averaging will mask out the extremes and the PDFs would have been smoothened. Secondly, the cloud condition varies significantly over short time scales, therefore, quantification at high temporal resolution is required. Last but not least, the previous study suggests that the $\omega 500$ is sensitive to local dynamics and subject to significant biases at the instantaneous scale (Trenberth et. cl, 2000). Research shows that the $\omega 500$ data with shorter time scales are unreliable (Höjgård et al., 2020). The monthly vertical motion can represent a mixture of ascending and descending atmospheric conditions. It is worth mentioning that by adopting the monthly mean of $\omega 500$ in our evaluation, the fluctuations of shorter time scales, where small-scale convection probably dominates, are ignored.